# Ubiquitination-mediated Golgi-to-endosome sorting determines the toxin-antidote duality of fission yeast *wtf* meiotic drivers

Jin-Xin Zheng[1,2], Tong-Yang Du[1,3], Guang-Can Shao[1], Zhu-Hui Ma[1], Zhao-Di Jiang [1], Wen Hu [1], Fang Suo[1], Wanzhong He [1], Meng-Qiu Dong [1,4] & Li-Lin Du [1,4] ✉

Killer meiotic drivers (KMDs) skew allele transmission in their favor by killing meiotic progeny not inheriting the driver allele. Despite their widespread presence in eukaryotes, the molecular mechanisms behind their selfish behavior are poorly understood. In several fission yeast species, single-gene KMDs belonging to the *wtf* gene family exert selfish killing by expressing a toxin and an antidote through alternative transcription initiation. Here we investigate how the toxin and antidote products of a *wtf*-family KMD gene can act antagonistically. Both the toxin and the antidote are multi-transmembrane proteins, differing only in their N-terminal cytosolic tails. We find that the antidote employs PY motifs (Leu/Pro-Pro-X-Tyr) in its N-terminal cytosolic tail to bind Rsp5/NEDD4 family ubiquitin ligases, which ubiquitinate the antidote. Mutating PY motifs or attaching a deubiquitinating enzyme transforms the antidote into a toxic protein. Ubiquitination promotes the transport of the antidote from the trans-Golgi network to the endosome, thereby preventing it from causing toxicity. A physical interaction between the antidote and the toxin enables the ubiquitinated antidote to translocate the toxin to the endosome and neutralize its toxicity. We propose that post-translational modification-mediated protein localization and/or activity changes may be a common mechanism governing the antagonistic duality of single-gene KMDs.

The killer meiotic drivers (KMDs), which have been found in animals, fungi, and plants, are selfish genetic elements that subvert Mendel's law of segregation by disabling meiotic progeny lacking the driver element[1]. A commonly employed mode of action of KMDs is the toxin-antidote mechanism[1,2]. The protein products of a toxin-antidote KMD form a toxin-antidote pair. The toxin exerts a killing activity towards all meiotic progeny and the antidote confers a protection against the toxin in progeny carrying the driver allele. Interestingly, the toxin and the antidote can be encoded by the same gene. Single-gene KMDs known or proposed to employ the toxin-antidote mechanism include

*wtf* genes of fission yeasts[3–9], the *Spk-1* gene of *Neurospora*[10], and the *Spok* family genes found in *Podospora* and other filamentous fungi[11–15]. How a single-gene KMD can produce proteins with antagonistic activities has been a perplexing mystery.

*wtf* (for with Tf LTRs) genes, which encode multi-transmembrane proteins, constitute the largest gene family in the fission yeast *Schizosaccharomyces pombe*[8,9,16]. Some of the *S. pombe wtf* genes are drivers that can express both a toxin and an antidote through alternative transcription initiation, whereas others express only an antidote[5,8,9]. The toxin and the antidote expressed by a *wtf* driver gene have

[1]National Institute of Biological Sciences, Beijing 102206, China. [2]Graduate School of Peking Union Medical College and Chinese Academy of Medical Sciences, Beijing 100730, China. [3]College of Life Sciences, Beijing Normal University, Beijing 100875, China. [4]Tsinghua Institute of Multidisciplinary Biomedical Research, Tsinghua University, Beijing 102206, China. ✉e-mail: dulilin@nibs.ac.cn

identical transmembrane domains, differing only in their N-terminal tails, where the antidote features an extension at its N-terminus. Presumably, this antidote-specific region plays a key role in determining the distinction between the toxin and the antidote. However, no activities have been assigned to the antidote-specific region.

Here, we show that the antidote-specific region is a binding platform for ubiquitin ligases and mediates the ubiquitination of the antidote. Ubiquitination directs the transport of the antidote from the trans-Golgi network (TGN) to the endosome, preventing the antidote from causing toxicity. Furthermore, through a physical interaction between the antidote and the toxin, the ubiquitinated antidote relocalizes the toxin to the endosome and thereby detoxifies the toxin. This ubiquitination-mediated toxicity neutralization mechanism is conserved among *wtf* genes, including *S. pombe wtf* genes encoding only antidotes and *wtf* genes present in other fission yeast species that

diverged from *S. pombe* about 100 million years ago. Our findings provide new insights into the molecular mechanisms underlying the actions of KMDs.

## Results

### The toxin and antidote products of *cw9* are active in vegetative cells

We used the *S. pombe wtf* gene *cw9*, which is an active KMD from the natural isolate CBS5557[9], as a model to study the molecular mechanism underlying the distinction between the toxin and the antidote. We will refer to its toxin as Cw9t and its antidote as Cw9a. Cw9a is 52 amino acids longer than Cw9t at the N terminus (Fig. 1a). Transcripts encoding Cw9a and Cw9t are initiated from upstream of coding exon 1 and within intron 1, respectively. As shown before[9], the removal of the sequence upstream of exon 1 resulted in a form of *cw9* that

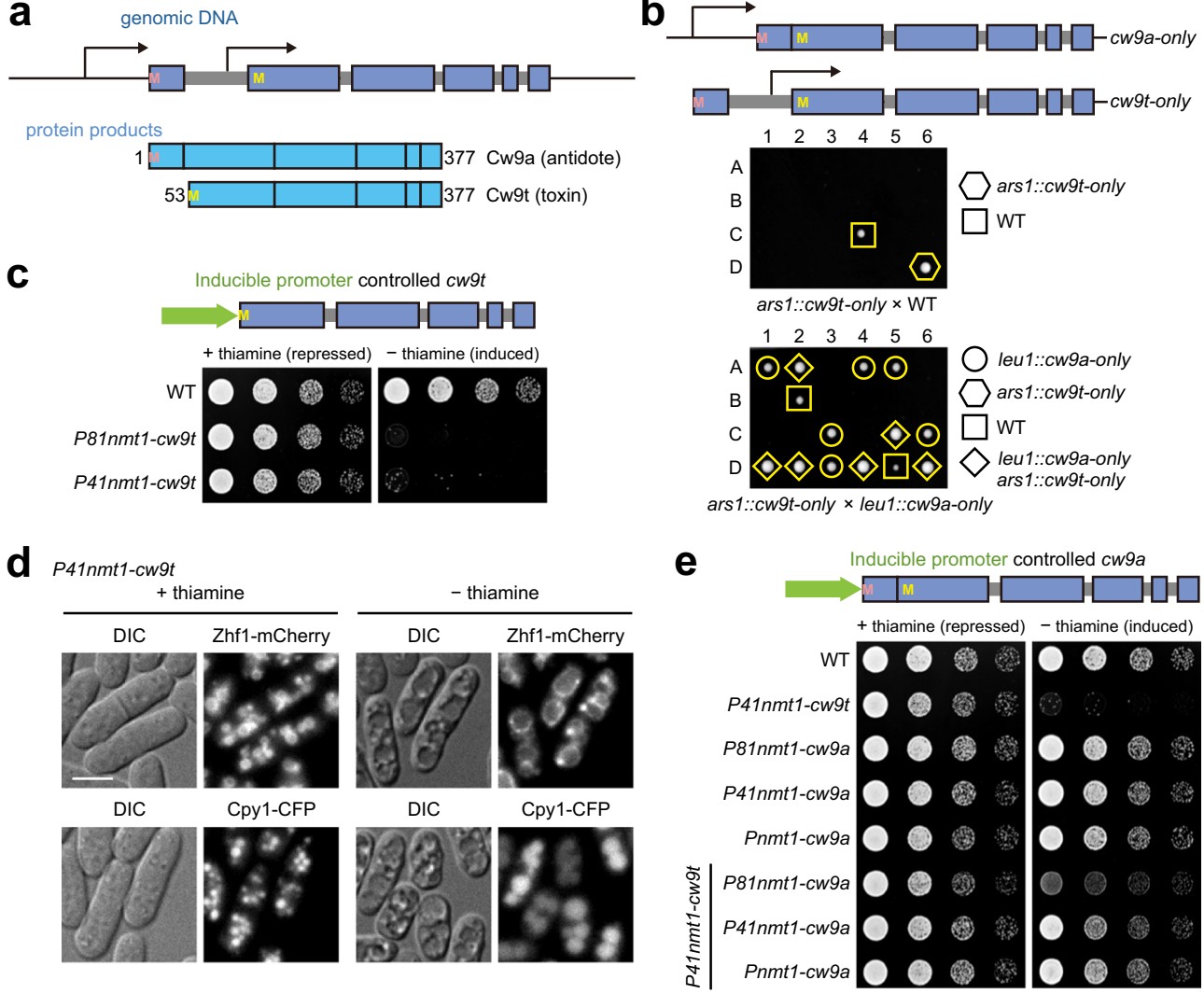

**Fig. 1 | Both the poison product and the antidote product of *cw9* are active in vegetative *S. pombe* cells. a** Diagrams depicting the two isoforms encoded by *cw9*. For the diagram of the genomic DNA (top), coding exons are denoted by boxes and introns are denoted by thick gray lines. Transcriptional start sites are shown as bent arrows. Start codons are denoted by the letter M. For the diagram of the protein products (bottom), regions corresponding to the exons are indicated by boxes and the numbering of amino acids is based on the sequence of Cw9a. **b** Tetrad analysis showed that a *cw9t-only* transgene inserted at the *ars1* locus on chromosome I killed nearly all spores in a cross to a wild-type (WT)

laboratory strain that lacks *cw9* and a *cw9a-only* transgene inserted at the *leu1* locus on chromosome II neutralized the spore killing effect of *cw9t-only*. The four spores of each tetrad are labeled A, B, C, and D. **c** Cw9t expressed in vegetative *S. pombe* cells from inducible promoters caused toxicity. **d** Expressing Cw9t in vegetative *S. pombe* cells from an inducible promoter caused vacuole enlargement. Zhf1 is a vacuole membrane marker. Cpy1 is a vacuole lumen marker. DIC, differential interference contrast. Bar, 5 μm. This experiment was repeated independently two times with similar results. **e** Cw9a expressed in vegetative *S. pombe* cells from inducible promoters neutralized the toxicity of Cw9t in a dose-dependent manner.

indiscriminately kills spore progeny of a cross between a strain carrying this form of *cw9* and a strain without *cw9*, presumably because only Cw9t but not Cw9a can be expressed (Fig. 1b). On the other hand, a Cw9a-only form created through deleting intron 1 was able to protect progeny from the killing effect of Cw9t expressed from a different locus (Fig. 1b). Thus, Cw9t and Cw9a can act as toxin and antidote respectively when they are not expressed from the same locus.

It has been shown that both the toxin and antidote products of the *Sk wtf4* gene from the *S. pombe var. kambucha* strain are active in vegetative cells[6]. To test whether the toxicity of Cw9t can manifest in vegetative *S. pombe* cells, we placed its coding sequence under the control of thiamine-repressible promoters (*Pnmt1*, *P41nmt1*, or *P81nmt1*, from the strongest to the weakest) (Fig. 1c)[17]. We found that Cw9t expressed from the *P81nmt1* or *P41nmt1* promoter prevented the growth of vegetative cells under the induction condition (minus thiamine) (Fig. 1c). We failed to obtain strains expressing Cw9t from the strongest *Pnmt1* promoter, probably due to toxicity caused by a higher level of basal expression. Cw9t expressed from the anhydrotetracycline (ahTet)-inducible promoter *PtetO7* also showed strong toxicity to vegetative cells under the induction condition (Supplementary Fig. 1a)[18]. Thus, Cw9t is a toxin that can kill not only *S. pombe* spores, but also *S. pombe* vegetative cells.

When examining cells using differential interference contrast (DIC) microscopy, we found that Cw9t induction in vegetative cells resulted in the appearance of large crater-like structures (Fig. 1d and Supplementary Fig. 1b). We suspected that these structures may correspond to enlarged vacuoles. Indeed, Zhf1, a vacuole membrane marker, localized at the boundaries of the structures, and Cpy1, a vacuole lumen marker, localized in the interior of the structures (Fig. 1d). Enlarged vacuoles can also be observed by electron microscopy analysis (Supplementary Fig. 1c). The reason behind this phenotype is unclear, but it serves as a convenient readout of the toxicity of Cw9t in vegetative *S. pombe* cells.

To test whether Cw9a can act in vegetative cells, we placed its coding sequence (without intron 1) under the control of thiamine-repressible promoters (Fig. 1e). Inducing the expression of Cw9a had no effect on vegetative cell growth, even when using the strongest *Pnmt1* promoter. When we co-expressed Cw9a and Cw9t, the toxicity of Cw9t expressed from the *P41nmt1* promoter was completely abrogated by Cw9a expressed from either the *P41nmt1* promoter or the *Pnmt1* promoter and strongly but incompletely mitigated by Cw9a expressed from the weakest *P81nmt1* promoter (Fig. 1e), indicating that the protecting activity conferred by Cw9a is dose-dependent. When Cw9t was expressed from the *P81nmt1* promoter, co-expressing Cw9a from even the *P81nmt1* promoter can abolish the toxicity (Supplementary Fig. 1d).

We also tested whether Cw9t and Cw9a are active in the vegetative cells of the budding yeast *Saccharomyces cerevisiae*. Similar to the observations made on the products of *Sk wtf4*[6], we found that Cw9t was toxic to vegetative budding yeast cells, and Cw9a can neutralize the toxicity of Cw9t in a dose-dependent manner (Supplementary Fig. 1e, f). Together, our results indicate that the activities of Cw9t and Cw9a are neither specific to spores nor limited to *S. pombe*.

### The antidote-specific region binds Rsp5/NEDD4 family ubiquitin ligases

Cw9a is predicted to be a multi-transmembrane protein with both its N terminus and C terminus in the cytosol (Supplementary Fig. 1g)[9]. The N-terminal cytosolic tail of Cw9a is predicted to be 105 amino acids long, with its most N-terminal 52 amino acids being absent in Cw9t (Fig. 2a). How does the presence or absence of this antidote-specific region, Cw9a(1-52), determine whether the protein is an antidote or a toxin? As Cw9a is essentially a combination of Cw9a(1-52) and Cw9t, we first addressed the question of how the presence of Cw9a(1-52) in the same polypeptide with Cw9t prevents the manifestation of the

toxicity of Cw9t. We hypothesized that Cw9a(1-52) may possess an activity that can neutralize the toxicity of Cw9t as long as it is physically associated with Cw9t. To test this idea, we artificially tethered Cw9a(1-52) to Cw9t using the noncovalent interaction between GFP and GBP (GFP binding protein)[19,20]. We co-expressed a GFP-tagged form of Cw9t, which is toxic to *S. pombe* vegetative cells (Fig. 2b), with Cw9a(1-52) or GBP-fused Cw9a(1-52). Cw9a(1-52) did not obviously ameliorate the toxicity of GFP-Cw9t, whereas Cw9a(1-52)-GBP abolished the toxicity of GFP-Cw9t (Fig. 2b), supporting the idea that Cw9a(1-52) possesses a toxicity-neutralizing activity that can detoxify associated Cw9t.

To explore the molecular mechanism underlying the toxicity-neutralizing activity of Cw9a(1-52), we searched for physical interactors of Cw9a(1-52) using affinity purification coupled with mass spectrometry (AP-MS) (Supplementary Data 1). One hit of the AP-MS analysis is Pub1 (Fig. 2c), an Rsp5/NEDD4 family ubiquitin ligase[21]. In *S. pombe*, there are three Rsp5/NEDD4 family ubiquitin ligases, Pub1, Pub2, and Pub3[22]. Pub1 and Pub3 are more similar to each other than to Pub2 and are redundantly essential for cell growth[22]. Pub1 has the highest expression level among the three[23,24]. We used a glutathione *S*-transferase (GST) pulldown assay to examine whether Cw9a(1-52) interacts with these three ubiquitin ligases and found that both Pub1 and Pub3 exhibited robust interactions with Cw9a(1-52), whereas Pub2 showed a much weaker interaction with Cw9a(1-52) (Fig. 2d). Similar results were obtained using an in vivo co-immunoprecipitation assay (Supplementary Fig. 1h), further validating the interactions.

The ability of Cw9a(1-52) to interact with Rsp5/NEDD4 family ubiquitin ligases led us to hypothesize that Cw9a(1-52) may neutralize the toxicity of associated Cw9t through recruiting ubiquitin ligases. This hypothesis predicts that the toxicity of Cw9t may be inhibited if a ubiquitin ligase is artificially tethered to it. To test this idea, we employed the GFP–GBP interaction again. Indeed, tethering Pub1 to Cw9t abolished the toxicity of Cw9t (Fig. 2e). As a control, tethering Pub1-C735S, a catalytically dead mutant of Pub1[25], to Cw9t did not affect the toxicity, indicating that the ubiquitin ligase activity is required for neutralizing the toxicity. Artificially tethering Pub3 to Cw9t can also abolish the toxicity, whereas tethering Pub2 only partially neutralized the toxicity (Fig. 2f). These results suggest that the antidote-specific region of Cw9a acts as a binding platform for Rsp5/NEDD4 family ubiquitin ligases, especially Pub1 and Pub3, and these ubiquitin ligases, when recruited near Cw9t, can neutralize the toxicity of Cw9t through promoting ubiquitination.

### PY motifs in the N-terminal tail of Cw9a are responsible for binding ubiquitin ligases and preventing Cw9a from becoming toxic

The substrate proteins of Rsp5/NEDD4 family ubiquitin ligases often contain one or multiple Leu/Pro-Pro-X-Tyr sequences (PY motifs), which are binding sites of the WW domains in the Rsp5/NEDD4 family ubiquitin ligases[26–29]. There are three PY motifs in Cw9a, at positions 30-33, 41-44, and 69-72, respectively (Fig. 3a). We named them PY1, PY2, and PY3. PY1 and PY2 are located within the antidote-specific region. In a sequence alignment of the N-terminal cytosolic tails of the antidote products of two active *wtf* driver genes from CBS5557 (*cw9* and *cw27*) and 16 non-pseudo *wtf* genes in the *S. pombe* reference genome, PY1 and PY2 are conserved in the products of all but the four most divergent *wtf* genes (*wtf7*, *wtf11*, *wtf14*, and *wtf15*), which do not have KMD activities[5], whereas PY3 is only present in the products of four genes, *cw9*, *cw27*, *wtf19*, and *wtf23* (Fig. 3a). The absence of PY3 in the product of the known active driver gene *wtf13* suggests that PY3 is not essential for the selfish actions[7].

To examine whether the PY motifs in the N-terminal cytosolic tail of Cw9a mediate interactions with Rsp5/NEDD4 family ubiquitin ligases, we performed GST pulldown analysis using *E. coli*-expressed Cw9a N-terminal fragments fused with GST and *E. coli*-expressed Pub1

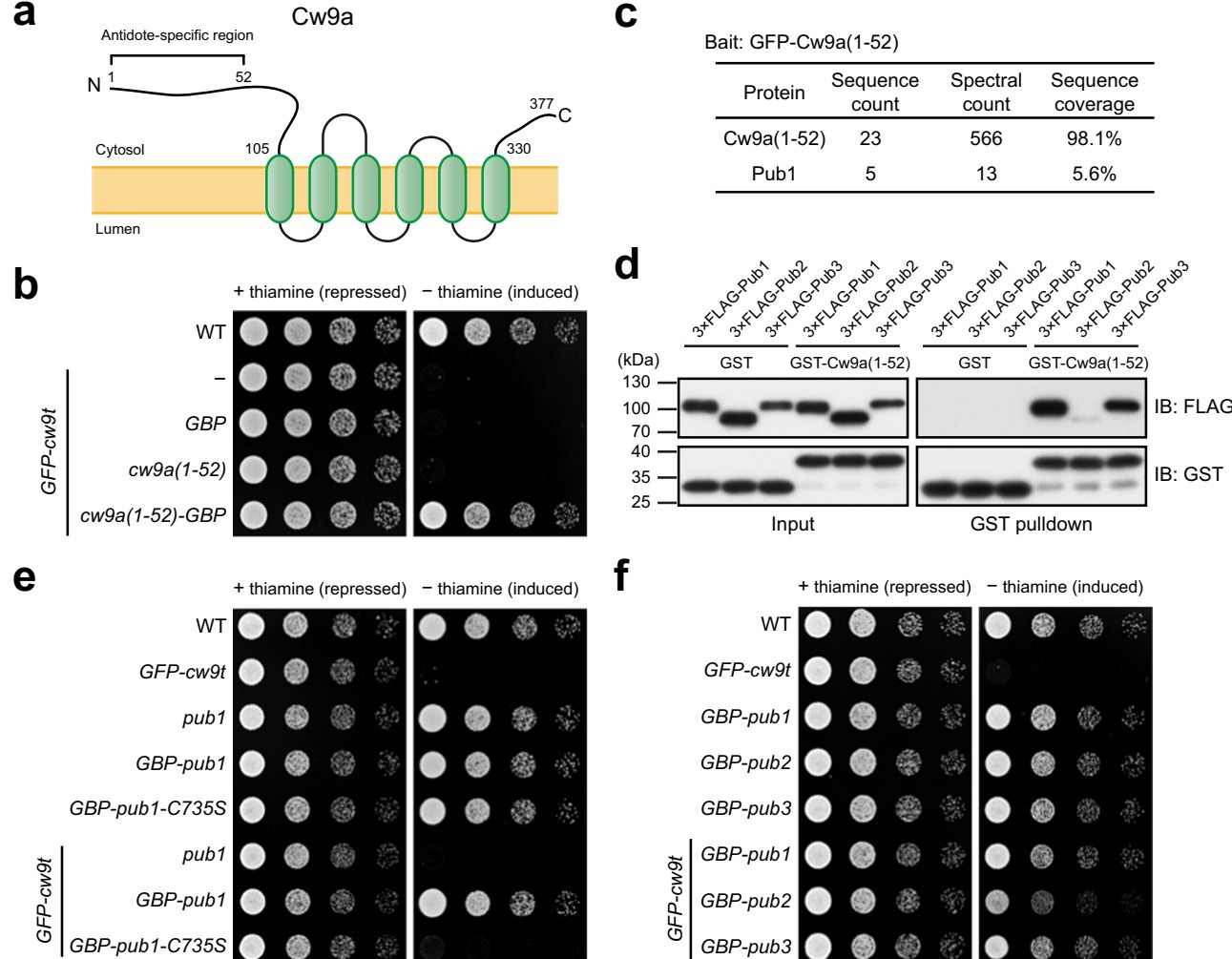

**Fig. 2 | The antidote-specific region of Cw9a functions as a binding module for the Rsp5/NEDD4 family ubiquitin ligases Pub1 and Pub3. a** A diagram depicting the TOPCONS-predicted membrane topology of Cw9a (see Supplementary Fig. 1g). The antidote-specific region is indicated by a bracket. **b** The antidote-specific region of Cw9a, Cw9a(1-52), when tethered to Cw9t through the interaction between GFP and the GFP-binding protein (GBP), neutralized the toxicity of Cw9t. GFP-Cw9t was expressed from the *P81nmt1* promoter. GBP, Cw9a(1-52), and Cw9a(1-52)-GBP were expressed from the *P41nmt1* promoter. **c** Pub1 co-purified with GFP-Cw9a(1-52). GFP-Cw9a(1-52) was purified using GFP-Trap beads. Co-purified proteins were identified by mass spectrometry. **d** GST pulldown assay showed that Pub1 and Pub3 but not Pub2 can strongly interact with Cw9a(1-52). The

lysate of *S. pombe* cells expressing 3×FLAG-tagged Pub1, Pub2, or Pub3 was mixed with the lysate of *E. coli* cells expressing GST or GST-tagged Cw9a(1-52) and pull-down was performed using glutathione beads. This experiment was repeated independently two times with similar results. **e** Artificial tethering of Pub1 to Cw9t neutralized the toxicity of Cw9t. GFP-Cw9t was expressed from the *P81nmt1* promoter. GBP-Pub1 was expressed from the *P41nmt1* promoter. Pub1-C735S is a catalytically inactive mutant of Pub1. **f** Tethering Pub1 or Pub3 to Cw9t confers a strong toxicity neutralization, whereas tethering Pub2 confers a weak toxicity neutralization. GFP-Cw9t was expressed from the *P81nmt1* promoter. GBP-tagged Pub proteins were expressed from the *P41nmt1* promoter. Source data are provided as a Source Data file.

lacking the N-terminal lipid-binding C2 domain but containing all three PY-motif-binding WW domains (Pub1-ΔC2) (Fig. 3b). The removal of the C2 domain has been shown to reduce the aggregation tendency of recombinant Rsp5 protein, while not impacting its ability to ubiquitinate PY-containing substrates[30]. It is known that mutating the strictly conserved tyrosine residue in a PY motif to alanine disrupts the ability of the PY motif to bind WW domains[31,32]. Thus, we introduced tyrosine-to-alanine mutations (denoted by asterisks) into the PY motifs of Cw9a. Mutating the three PY motifs individually (PY1*, PY2*, or PY3*) did not obviously affect the ability of the N-terminal cytosolic tail of Cw9a, Cw9a(1-104) (abbreviated as N in Fig. 3b), to bind Pub1-ΔC2. When two out of the three PY motifs were mutated simultaneously, the PY1*PY2* and PY2*PY3* combinations, but not the PY1*PY3* combination, resulted in dramatically reduced but still detectable interactions. Disrupting all three PY motifs (3PY*) weakened the interaction to an undetectable level. Consistently, Cw9a(24-52), a 29-amino-acid

fragment containing both PY1 and PY2, interacted with Pub1-ΔC2 as strongly as Cw9a(1-104). Cw9a(53-104), which contains only PY3, interacted weakly with Pub1-ΔC2, whereas Cw9a(1-23) that lacks any PY motifs did not interact with Pub1-ΔC2 (Fig. 3b). Similar results were obtained when using Pub3 as prey to perform GST pulldown assay (Supplementary Fig. 2a). Together, these results indicate that the three PY motifs of Cw9a act redundantly to promote interactions with Pub1 and Pub3. PY2 is sufficient to mediate a strong interaction by itself, whereas PY1 and PY3 individually are much weaker binding motifs but together can mediate a robust interaction.

To examine the in vivo roles of the PY motifs, PY-mutated forms of Cw9a were expressed in vegetative *S. pombe* cells using the inducible *P81nmt1* promoter. Mutations that did not substantially affect Pub1/Pub3 binding in the GST pulldown analysis, including PY1*, PY2*, PY3*, and PY1*PY3*, did not alter the non-toxic nature of Cw9a (Fig. 3c and Supplementary Fig. 2b). In contrast, mutations

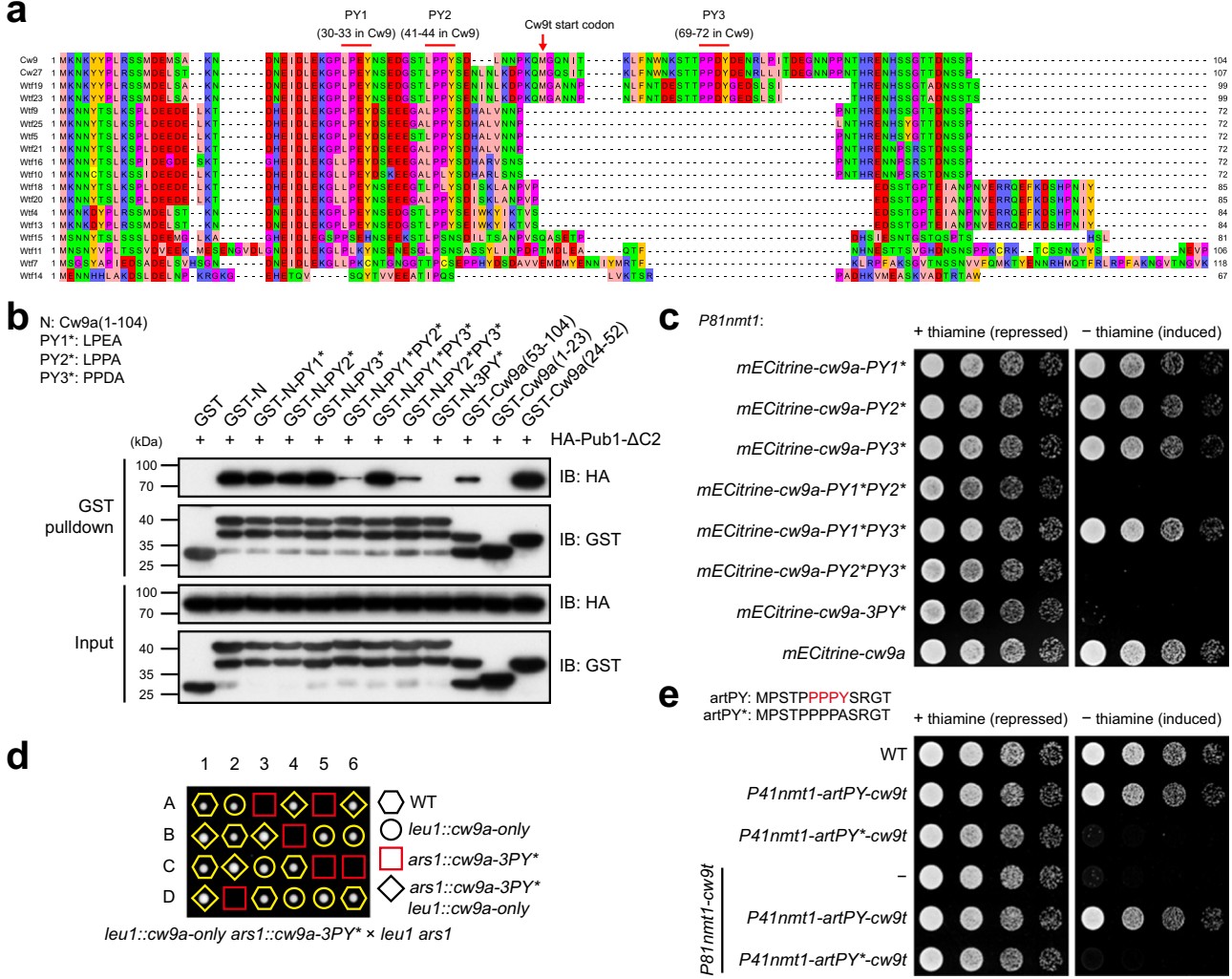

**Fig. 3 | PY motifs in the N-terminal cytosolic tail of Cw9a mediate Pub1 binding and prevent Cw9a from becoming toxic. a** Multiple sequence alignment of the N-terminal cytosolic tails of the antidote products of 18 *wtf* genes. Three PY motifs in Cw9a are highlighted. **b** GST pulldown assay showed that PY motifs in the N-terminal cytosolic tail of Cw9a are redundantly required for Pub1 binding. Lysates of *E. coli* cells expressing GST-tagged Cw9a N-terminal fragments were mixed with recombinant 6xHis-HA-Pub1-ΔC2 and pulldown was performed using glutathione beads. N denotes Cw9a(1-104). Asterisk denotes the Y-to-A mutation in the PY motif. Based on the observed molecular weights, the top band in lanes containing multiple bands probably corresponds to intact GST fusion proteins, while the lower bands are likely the result of proteolytic cleavage events that occurred during the preparation of the recombinant proteins. This experiment was repeated independently two times with similar results. **c** PY motif mutations strongly diminishing Pub1 binding rendered Cw9a toxic to vegetative cells. **d** Cw9a-3PY* under the control of the native promoter caused self-killing during sexual reproduction. **e** Fusing an artificial PY motif (artPY) to Cw9t abolished the toxicity of Cw9t and converted Cw9t to an antidote. Source data are provided as a Source Data file.

that markedly weakened ubiquitin ligase binding in the pulldown assay, including PY1*PY2*, PY2*PY3*, and 3PY*, rendered Cw9a strongly toxic to vegetative cells (Fig. 3c). Thus, PY motif-mediated binding to Rsp5/NEDD4 family ubiquitin ligases keeps Cw9a in a non-toxic state.

We found that the 3PY* mutation can also convert Cw9a from non-toxic to toxic in vegetative *S. cerevisiae* cells (Supplementary Fig. 2c), indicating that PY motifs in Cw9a can mediate toxicity neutralization in *S. cerevisiae*, probably because PY motif binding is a conserved feature of Rsp5/NEDD4 family ubiquitin ligases, which are ubiquitously present in fungi and animals[33].

To examine whether PY motifs are also important for preventing Cw9a from becoming toxic during sexual reproduction, we placed Cw9a-3PY* under the control of the native upstream sequence. This form, referred to as *cw9a-3PY**, can only be introduced into a strain that also contains a wild-type form of *cw9a*, presumably because of the toxicity of Cw9a-3PY* to vegetative cells. When the strain containing both *cw9a-3PY** and *cw9a* was crossed to a strain without any forms of

*cw9*, the resulting spores that contained only *cw9a-3PY** were inviable, while the viability of other spores was normal (Fig. 3d). Thus, *cw9a-3PY** behaved as a suicidal gene during sexual reproduction, indicating that PY motif-mediated ubiquitin ligase binding is required to prevent the antidote from becoming a toxin in the native context of KMD actions.

PY3 is located downstream of the antidote-specific region and thus is present in Cw9t. The pulldown analyses showed that PY3 can support a weak binding with Pub1 and Pub3 (Fig. 3b and Supplementary Fig. 2a). To determine whether the presence of PY3 weakens the toxicity of Cw9t, we expressed Cw9t in vegetative cells using an attenuated version of the *P81nmt1* promoter (*P81nmt1a*) so that cell growth was inhibited but not completely prevented. We found that mutating PY3 of Cw9t did not alter the extent of growth inhibition (Supplementary Fig. 2d), indicating that PY3 does not influence the toxicity of Cw9t, perhaps because PY3-mediated ubiquitin ligase binding is too weak to reach a threshold needed to exert a neutralizing effect on the toxicity.

To further ascertain the toxicity-neutralizing role of PY motifs, we tested whether simply adding a strong exogenous PY motif to Cw9t is sufficient to neutralize its toxicity. We fused to the N-terminus of Cw9t a PY-motif-containing artificial peptide (artPY, MPSTPPPPYSRGT), which is known to promote strong in vitro ubiquitination by the Rsp5 ubiquitin ligase[30]. This fusion abolished the toxicity of Cw9t (Fig. 3e). As a control, fusing a PY-mutated peptide (artPY*, MPSTPPPPASRGT) to Cw9t did not affect the toxicity. When artPY-Cw9t and Cw9t were co-expressed, artPY-Cw9t behaved like an antidote and neutralized the toxicity of Cw9t (Fig. 3e). Thus, the antidote-specific region of Cw9a can be substituted by an exogenous PY motif, indicating that the main function of the antidote-specific region is binding Rsp5/NEDD4 family ubiquitin ligases through PY motifs.

### Ubiquitination sites in the N-terminal cytosolic tail of Cw9a are necessary for preventing Cw9a from becoming toxic

Because the N-terminal cytosolic tail of Cw9a interacts with Rsp5/NEDD4 family ubiquitin ligases, we tested whether this region of Cw9a harbors ubiquitination sites that can be modified by Rsp5/NEDD4 family ubiquitin ligases. We performed in vitro ubiquitination reactions using recombinant Cw9a(1-104) as the substrate and recombinant Pub1-ΔC2 as the ubiquitin ligase (E3) (Fig. 4a). Pub1-ΔC2 lacks the N-terminal lipid-binding C2 domain but contains the catalytic domain and all three WW domains. Upon SDS-PAGE separation of the reaction mixes, we observed that the unmodified form of Cw9a(1-104) decreased and higher molecular weight forms of Cw9a(1-104) appeared when and only when ubiquitin, the ubiquitin-activating enzyme (E1), the ubiquitin-conjugating enzyme (E2), and E3 were all present, indicative of ubiquitination of Cw9a(1-104). The extent of ubiquitination correlated with the amount of E3. Thus, Cw9a(1-104) is a ubiquitination substrate of Pub1.

To test whether PY motifs are important for the ubiquitination of Cw9a(1-104) by Pub1, we performed ubiquitination reactions on PY-mutated Cw9a(1-104) proteins (Fig. 4b). Mutating the three PY motifs individually caused mild reduction of ubiquitination, with PY2* showing the strongest effect. Among the double PY mutations, PY1*PY2* and PY2*PY3* but not PY1*PY3* mutations caused substantially more severe reduction of ubiquitination than PY2*. 3PY* mutations reduced the ubiquitination to an undetectable level. These results parallel the binding assay results shown earlier, and indicate that PY motifs are needed for the ubiquitination of Cw9a(1-104) by Pub1.

To obtain evidence on the in vivo role of Cw9a ubiquitination, we tested whether artificially preventing the ubiquitination of Cw9a can render it toxic. We tethered UL36, a deubiquitinating enzyme (DUB), to Cw9a using the GFP–GBP interaction[34–37]. Cw9a fused at the N terminus with mECitrine, a variant of GFP, was expressed from the thiamine-repressible P81nmt1 promoter and DUB fused with GBP was expressed from the ahTet-inducible PtetO7 promoter. The co-expression of GBP-DUB, but not the catalytically inactive GBP-DUB*, with mECitrine-Cw9a caused toxicity to vegetative S. pombe cells (Fig. 4c), indicating that ubiquitination of Cw9a is essential for preventing it from becoming toxic.

There are a total of 16 lysines in the cytosolic tails and loops of Cw9a (Fig. 4d). We predicted that mutating all these lysines should prevent Cw9a from being ubiquitinated and render Cw9a toxic. Indeed, when all 16 lysines were mutated to arginines, the resulting mutant, Cw9a-16R, was toxic to vegetative cells (Fig. 4e). Next, to define a minimal set of ubiquitination sites important for toxicity neutralization, we mutated only the lysines in the N-terminal cytosolic tail. There are seven lysines in this region (Fig. 4d, f). Mutating the five lysines in the antidote-specific region (Cw9a-5R) or mutating the two lysines in the remaining portion of the N-terminal cytosolic tail (Cw9a-2R) had no effect, whereas mutating all seven lysines (Cw9a-7R) rendered Cw9a toxic (Fig. 4f). Reverting any one of the seven mutations back to lysine abolished the toxicity of Cw9a-7R (Fig. 4g), suggesting

that these seven lysines can all be ubiquitinated and the ubiquitination of these lysines is redundantly required for toxicity neutralization. Consistently, using mass spectrometry analysis, we found that all seven lysines in the N-terminal cytosolic tail can be ubiquitinated in vitro by Pub1 or Pub3 (Fig. 4h and Supplementary Fig. 2e). Together, these results indicate that PY motif-dependent ubiquitination of the N-terminal cytosolic tail of Cw9a is necessary for preventing Cw9a from becoming toxic.

### The vacuolar targeting of Cw9a is mediated by the ESCRT machinery

It has been reported that the antidote and the toxin products of the Sk wtf4 gene from the S. pombe var. kambucha strain exhibited distinct subcellular localization patterns, with the antidote protein localizing to vacuoles and the toxin protein forming cytoplasmic puncta[6,8]. To examine the localization of the protein products of cw9, we performed microscopy analysis on Cw9t and Cw9a tagged with fluorescent proteins. Cw9t was tagged by inserting a fluorescent protein at an internal position of the N-terminal cytosolic tail (between S96 and G97, amino acid numbering based on the sequence of Cw9a), because this way of tagging maintained the spore killing activity of Cw9t better than C-terminal tagging or N-terminal tagging (Supplementary Fig. 3a). We refer to Cw9t tagged internally with GFP as GFP^int-Cw9t. Cw9a was tagged at the N terminus, as N-terminal tagging did not obviously affect the antidote activity of Cw9a (Supplementary Fig. 3b). Similar to the observations made on the protein products of Sk wtf4[6,8], GFP-Cw9a showed perfect co-localization with the vacuole lumen marker Cpy1-CFP in spores (Supplementary Fig. 3c), whereas GFP^int-Cw9t formed puncta not overlapping with vacuoles in spores (Supplementary Fig. 3d).

In vegetative cells, Cw9t and Cw9a tagged with the yellow fluorescent protein mECitrine and expressed from the P81nmt1 or P41nmt1 promoter behaved like the untagged proteins in terms of the toxicity of Cw9t and the antidote activity of Cw9a (Supplementary Fig. 3e). Similar to the situation in spores, in vegetative cells, mECitrine^int-Cw9t formed cytoplasmic puncta and mECitrine-Cw9a localized to the vacuole lumen (Fig. 5a). mECitrine-Cw9a localized in the vacuole lumen is expected to be cleaved by vacuolar proteases to release free mECitrine, which is resistant to protease digestion[38]. Indeed, immunoblotting analysis showed that mECitrine-Cw9a was processed into free mECitrine in vegetative cells in a manner dependent on the two main vacuolar proteases Isp6 and Psp3 (Fig. 5b)[39].

It has been shown that the co-expression of Sk wtf4 antidote protein results in the relocalization of Sk wtf4 toxin protein to vacuoles[6]. Similarly, we found that the co-expression of mCherry-Cw9a altered the localization of mECitrine^int-Cw9t in vegetative S. pombe cells, resulting in a perfect co-localization of mECitrine^int-Cw9t with mCherry-Cw9a in the vacuole lumen (Fig. 5c). This effect of Cw9a on Cw9t is likely mediated by a physical interaction between Cw9a and Cw9t, as mCherry-Cw9a can be co-immunoprecipitated with mECitrine^int-Cw9t (Fig. 5d). Thus, as proposed before for Sk wtf4[6], Cw9a may exert its antidote activity through binding Cw9t and altering the localization of Cw9t.

Because mutating PY motifs in the N-terminal cytosolic tail of Cw9a can render Cw9a toxic, we hypothesized that PY-mutated forms of Cw9a that become toxic may have a localization pattern similar to that of Cw9t. Indeed, toxic Cw9a mutants, including PY1*PY2*, PY2*PY3*, and 3PY* mutants, no longer showed prominent vacuolar localization but instead mainly localized to cytoplasmic puncta, whereas non-toxic PY-mutated forms of Cw9a, including PY1*, PY2*, PY3*, and PY1*PY3*, still mainly localized to the vacuole lumen (Fig. 5e). Thus, the targeting of Cw9a to the vacuole lumen requires the PY motifs. The 3PY* mutant of Cw9a exhibited co-localization with Cw9t (Supplementary Fig. 3f), indicating that in the absence of PY motif-dependent ubiquitination, Cw9a localizes to the same compartment as Cw9t.

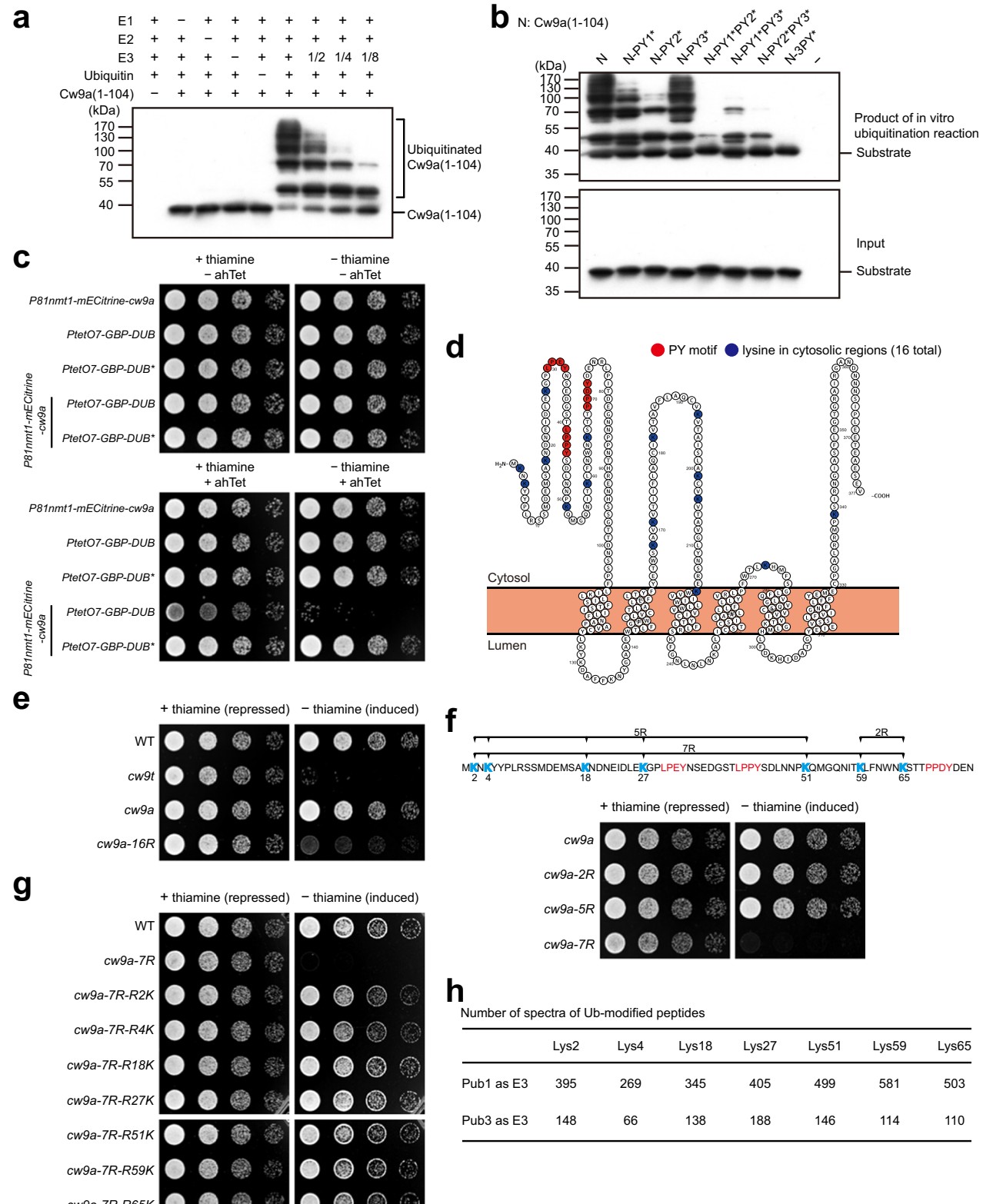

Ubiquitination of transmembrane proteins by Rsp5/NEDD4 family ubiquitin ligases is known to be a sorting signal recognized by the endosomal sorting complex required for transport (ESCRT) machinery[40]. The ESCRT machinery sorts ubiquitinated transmembrane proteins into intra-endosomal vesicles, which end up in the vacuole lumen upon endosome-vacuole fusion. As the PY motifs in Cw9a promote the ubiquitination of Cw9a by Rsp5/NEDD4 family ubiquitin ligases, we hypothesized that the vacuolar targeting of Cw9a may rely on the ESCRT machinery. Supporting this idea, we found that in the absence of Vps24 or Sst4, which are components of the ESCRT-III complex and the ESCRT-0 complex, respectively, Cw9a no longer localized to the vacuole lumen, and instead formed cytoplasmic puncta, which tend to be larger than the puncta formed by Cw9t (Fig. 5f and Supplementary Fig. 3g). The Cw9a puncta in ups24Δ cells showed substantial co-localization with two endosome markers: the ESCRT-0 complex subunit Hse1 and the endosomal membrane protein

**Fig. 4 | PY motif-dependent ubiquitination of the N-terminal cytosolic tail of Cw9a renders Cw9a non-toxic. a** Pub1-catalyzed in vitro ubiquitination of Cw9a(1-104). All components of the in vitro ubiquitination reactions were purified from *E. coli*. Immunoblotting was used to detect Cw9a(1-104), which was fused to a lysine-free 8×myc* tag that can be recognized by the myc antibody[62]. **b** PY motif mutations diminished the in vitro ubiquitination of Cw9a(1-104). **c** Tethering a de-ubiquitinating enzyme (DUB) UL36 to Cw9a through the mECitrine-GBP interaction rendered Cw9a toxic. UL36-C40S (DUB*) is a catalytically inactive mutant of UL36. **d** Diagram highlighting lysines in the cytosolic regions of Cw9a. Blue circles denote the 16 lysines residing in the cytosolic tails or loops, including K216, which is predicted to localize at the cytosolic edge of a transmembrane helix. Red circles denote the PY motifs. This diagram was generated using Protter[69]. **e** Mutating all 16 lysines in the cytosolic regions of Cw9a to arginines (Cw9a-16R) rendered Cw9a

toxic to vegetative cells. Expression was under the control of the *P41nmt1* promoter. **f** Mutating all 7 lysines in the N-terminal cytosolic tail of Cw9a (Cw9a-7R) rendered Cw9a toxic, whereas mutating two subsets of the 7 lysines (Cw9a-5R and Cw9a-2R) did not cause toxicity. Expression was under the control of the *P41nmt1* promoter. **g** The 7 lysines in the N-terminal cytosolic tail of Cw9a are redundantly required for preventing the toxicity. Mutated residues in the Cw9a-7R mutant were individually reverted to lysine. Expression was under the control of the *P41nmt1* promoter. **h** All seven lysines in the N-terminal cytosolic tail of Cw9a can be ubiquitinated in vitro by Pub1 and Pub3. In vitro ubiquitination of GST-Cw9a(1-104) was performed using either Pub1 or Pub3 as E3 and ubiquitination sites were identified by mass spectrometry. Experiments in (**a**, **b**) were repeated independently two times with similar results. Source data are provided as a Source Data file.

SPAC15A10.06 (ortholog of *S. cerevisiae* Nhx1) (Fig. 5f, g), suggesting that Cw9a accumulates on the endosome when the ESCRT machinery is defective. Thus, Cw9a is transported to the vacuole through the endosome and the ESCRT machinery is required for the endosome-to-vacuole transport of Cw9a. Further supporting this conclusion, we found that the vacuole lumen localization of Cw9a was unaffected in mutants defective in the AP3 pathway, an endosome-bypassing Golgi-to-vacuole trafficking pathway (Supplementary Fig. 3h).

The ubiquitination on the cargo transmembrane proteins transported by the ESCRT machinery is known to be removed by DUBs when the cargo proteins are sequestered into the endosome[41]. The transient nature of ubiquitination makes its in vivo detection challenging. An ESCRT mutant preventing the endosomal sequestration of cargo proteins, such as *vps24Δ*, causes the accumulation of ubiquitinated cargo proteins and makes their detection easier[42]. Using biotin-tagged ubiquitin (biotin-Ub) to enrich ubiquitinated proteins[36,43], we detected ubiquitinated Cw9a in *vps24Δ* cells (Fig. 5h). The PY1*PY3* mutation, which does not prevent the vacuolar localization of Cw9a in the wild-type background, diminished but did not abolish ubiquitination in *vps24Δ* cells, whereas PY1*PY2*, PY2*PY3*, and 3PY* mutations, which prevent the vacuolar localization of Cw9a in the wild-type background, decreased ubiquitination to an undetectable level in *vps24Δ* cells (Fig. 5h). These results support that PY motif-mediated ubiquitination determines the endosome-to-vacuole transport of Cw9a.

**Trafficking of Cw9a and Cw9t from the TGN to the endosome prevents their toxicity**

Although deleting *vps24* prevented the vacuolar targeting of Cw9a, Cw9a did not show toxicity in *vps24Δ* cells (Fig. 5i). This is not due to a suppression effect of *vps24Δ* on the manifestation of toxicity, as Cw9t and PY-mutated forms of Cw9a showed toxicity in *vps24Δ* cells (Supplementary Fig. 4a). Thus, ubiquitination-dependent neutralization of the toxicity of Cw9a still occurs when Cw9a localizes at the endosome. We predicted that in this situation, Cw9a can also act as an antidote to inhibit the toxicity of Cw9t. Indeed, mTurquoise2-Cw9a neutralized the toxicity of mECitrine^int^-Cw9t in *vps24Δ* cells (Fig. 5j). Moreover, they co-localized with the endosome marker Hse1 (Supplementary Fig. 4b). In contrast, Cw9t and Cw9a-3PY*, when individually expressed, did not show obvious co-localization with Hse1 in *vps24Δ* cells (Supplementary Fig. 4c). These results suggest that endosomal localization prevents Cw9a and Cw9t from becoming toxic.

To determine which compartment Cw9t localizes to in the absence of Cw9a, we performed co-localization analysis using a number of organelle markers exhibiting punctate patterns and found that mECitrine^int^-Cw9t puncta partially co-localized with the trans-Golgi network (TGN) marker Sec72-mCherry (Supplementary Fig. 4d). The TGN is a dynamic organelle and typical TGN-localizing proteins only exhibit partial overlap with each other[44]. We performed a time-lapse analysis and found that the cytoplasmic puncta of Cw9t exhibited dynamic behaviors and showed transient co-localization with Sec72 puncta (Fig. 5k). The most likely trafficking route that Cw9t takes to

reach the TGN is the secretary pathway starting at the endoplasmic reticulum (ER). Supporting this idea, overexpressing a dominant negative Sar1-T34N mutant, which impedes ER-to-Golgi transport at the ER exit step[45], caused mECitrine^int^-Cw9t to show a perfect co-localization with the ER marker Ost4-mCherry (Supplementary Fig. 4e). These results suggest that Cw9t enters the secretary pathway at the ER and is transported through the Golgi to reach the TGN.

We hypothesized that Cw9a follows the same trafficking route as Cw9t to reach the TGN, where the ubiquitination sorting signal on Cw9a directs its transport to the endosome. This hypothesis predicts that ubiquitin-binding trafficking factors may promote the TGN-to-endosome transport of Cw9a. The most obvious candidates for such factors are Golgi-localized gamma-ear-containing ARF-binding (GGA) family trafficking adapters that can bind ubiquitin and act in TGN-to-endosome transport[46,47]. There are two GGA-encoding genes in *S. pombe*, *gga21* and *gga22*, which play partially redundant roles[47]. Cw9a remained non-toxic in single deletion mutants lacking *gga21* or *gga22* (Supplementary Fig. 5a). However, in the *gga21Δ gga22Δ* double deletion mutant, which has a moderate growth defect, the expression of Cw9a further weakened the growth, indicative of a mild toxicity (Supplementary Fig. 5a).

Live cell imaging showed that in *gga21Δ gga22Δ* cells, Cw9a exhibited mainly a vacuole lumen localization, but a small subset of Cw9a signals were observed on cytoplasmic puncta outside of vacuoles (Supplementary Fig. 5b). Thus, the loss of Gga21 and Gga22 only weakly disturbed the trafficking of Cw9a, possibly because there are additional trafficking factors promoting the TGN-to-endosome trafficking of Cw9a. We surmised that partially reducing the ubiquitination of Cw9a using PY mutations that moderately weaken the in vitro ubiquitination of Cw9a(1-104) may enhance the phenotypic consequence of losing Gga21 and Gga22. Indeed, Cw9a-PY2* and Cw9a-PY1*PY3*, which localize to the vacuole lumen and do not show toxicity in the wild-type background, localized to cytoplasmic puncta and caused strong toxicity in the *gga21Δ gga22Δ* double deletion background (Supplementary Fig. 5b, c). Furthermore, their puncta showed partial overlap with the TGN marker Sec72-mCherry (Supplementary Fig. 5d). These findings suggest that ubiquitination-mediated trafficking of Cw9a from the TGN to the endosome renders it non-toxic. Further supporting this model, ubiquitination-defective Cw9a-3PY* protein exhibited co-localization with the TGN marker (Supplementary Fig. 5e).

To further investigate how the ubiquitinated Cw9a neutralizes the toxicity of Cw9t, we examined whether the detoxification of Cw9t requires its ubiquitination. Mutating all 11 cytosol-facing lysines of Cw9t to arginines did not affect its toxicity or the neutralization of its toxicity by Cw9a (Supplementary Fig. 5f), indicating that Cw9t ubiquitination is not necessary for detoxification. Given that Cw9a and Cw9t physically interact (Fig. 5d), we propose that Cw9a forms a complex with Cw9t, and this Cw9a-Cw9t complex is transported from the TGN to the endosome in a manner dependent on the PY motif-mediated ubiquitination of Cw9a, which serves as the sorting signal recognized by trafficking factors.

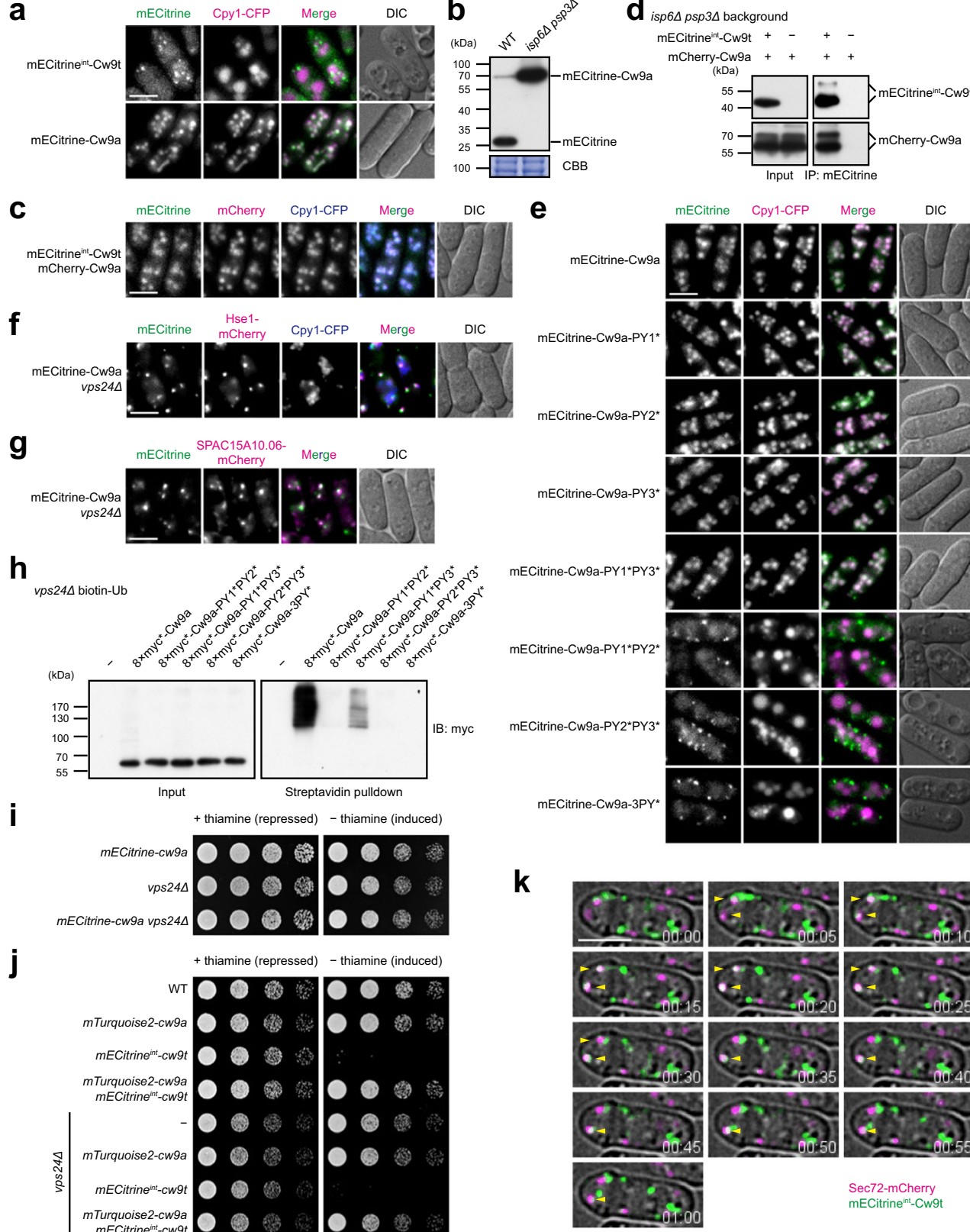

## Ubiquitination neutralizes the toxicity of antidotes encoded by other *wtf* genes from *S. pombe* and *S. octosporus*

Among the 16 non-pseudo *wtf* genes in the *S. pombe* reference genome[9], all but the four most divergent genes (*wtf7*, *wtf11*, *wtf14*, and *wtf15*) encode antidotes harboring the PY1 and PY2 motifs (Fig. 3a). Our analysis of Cw9a led us to hypothesize that like Cw9a,

these PY motif-containing antidote proteins may also be kept non-toxic by ubiquitination. This model predicts that artificially preventing ubiquitination of these antidote proteins may release their toxicity. To test this idea, we applied the DUB tethering analysis on the antidote products of these 16 non-pseudo *wtf* genes.

**Fig. 5 | PY motif-dependent ubiquitination of Cw9a promotes its trafficking from the TGN to the endosome and from the endosome to the vacuole. a** Cw9t localized to cytoplasmic puncta and Cw9a localized to the vacuole lumen in vegetative *S. pombe* cells. Bar, 5 µm. **b** mECitrine-Cw9a was processed to free mECitrine in wild-type cells but not in cells lacking vacuolar proteases Isp6 and Psp3. Post-blotting staining of the PVDF membrane using Coomassie brilliant blue (CBB) served as the load control. **c** Cw9t exhibited a vacuole lumen localization when co-expressed with Cw9a. Internally tagged Cw9t was expressed from the *P81nmt1* promoter. N-terminally tagged Cw9a was expressed from the *P41nmt1* promoter. Bar, 5 µm. **d** Cw9a was co-immunoprecipitated with Cw9t. Cw9t was internally tagged and Cw9a was N-terminally tagged. The *isp6Δ psp3Δ* background was used to ameliorate degradation. **e** PY mutations that rendered Cw9a toxic changed its localization to a Cw9t-like cytoplasmic punctate pattern. Bar, 5 µm. **f** Cw9a co-localized with the endosome marker Hse1 in *vps24Δ* cells. Bar, 5 µm.

**g** Cw9a co-localized with the endosome marker SPAC15A10.06 in *vps24Δ* cells. Bar, 5 µm. **h** In vivo ubiquitination of Cw9a in the *vps24Δ* background was diminished by PY mutations. Under a denaturing condition, biotin-tagged ubiquitin was precipitated using streptavidin beads and Cw9a in the precipitates was detected by immunoblotting. **i** Cw9a remained non-toxic in *vps24Δ* cells. *vps24Δ* itself causes a slight growth defect. Cw9a expression was under the control of the *P41nmt1* promoter. **j** Cw9a neutralized the toxicity of Cw9t in the *vps24Δ* background. Cw9a expression was under the control of the *P41nmt1* promoter and Cw9t expression was under the control of the *P81nmt1* promoter. **k** Time-lapse imaging showed that Cw9t exhibited transient co-localization with the TGN marker Sec72. Yellow arrowheads indicate overlapping signals of Cw9t and Sec72. Internally tagged Cw9t was expressed from the *P81nmt1* promoter. Bar, 5 µm. Experiments in (**a**–**h**) and (**k**) were repeated independently two times with similar results. Source data are provided as a Source Data file.

We crossed a strain expressing a GFP-tagged antidote protein from a thiamine-repressible promoter to a strain expressing GBP-DUB or a strain expressing GBP-DUB* (catalytically dead DUB) on mating/sporulation media containing thiamine and performed tetrad dissection analysis on media lacking thiamine. Spore progeny harboring both a *GFP-antidote* transgene and a *GBP-DUB* transgene were inviable for nine *wtf* genes, including all four genes that can express both the antidote and the toxin (*wtf4*, *wtf13*, *wtf19*, and *wtf23*) and five genes that can only express the antidote (*wtf9*, *wtf10*, *wtf16*, *wtf18*, and *wtf21*) (Fig. 6a). For the antidote-only gene *wtf25*, spore progeny of the *GFP-antidote GBP-DUB* genotype formed only extremely small colonies. Antidotes encoded by the other six non-pseudo *wtf* genes (*wtf5*, *wtf20*, and the four divergent genes) did not show toxicity upon DUB tethering (Supplementary Fig. 6a). For all 16 *wtf* genes, combining a *GFP-antidote* transgene with a *GBP-DUB*\* transgene did not result in spore viability loss or growth defect (Fig. 6a and Supplementary Fig. 6a). These results indicate that PY motif-containing antidotes mostly possess intrinsic toxicity that needs to be neutralized by ubiquitination to avoid suicidal killing.

The protein products of two antidote-only genes *wtf5* and *wtf10* are 82.9% identical in their amino acid sequences (Supplementary Fig. 6b). However, only Wtf10 but not Wtf5 displayed toxicity in the DUB tethering analysis. To understand the reason behind this difference, we performed sequence swapping between Wtf5 and Wtf10, and found that the difference is determined by the Wtf5(73-142) region (Supplementary Fig. 6c). Within this region, these two proteins only differ at two residues (Supplementary Fig. 6b). Replacing either residue in Wtf5 with the residue in Wtf10 rendered Wtf5 toxic in the DUB tethering analysis (Supplementary Fig. 6d). Thus, Wtf5 may have only recently lost its intrinsic toxicity and can regain the toxicity through a single point mutation.

*wtf* genes are present not only in *S. pombe* but also in four other fission yeast species: *S. octosporus*, *S. lindneri*, *S. osmophilus*, and *S. cryophilus*[4,48]. These four species share a common ancestor that lived ~30 million years ago, and that common ancestor diverged from *S. pombe* around 100 million years ago[48,49]. Since *S. lindneri* still lacks a high-quality reference genome, the full catalogs of *wtf* genes are only available for *S. octosporus*, *S. osmophilus*, and *S. cryophilus*[4,49]. Inspecting sequence alignments of proteins encoded by intact *wtf* genes in these three species showed that the N-terminal cytosolic tails of the antidote isoforms all contain a single conserved PY motif (Supplementary Fig. 7a). We predicted that, like the situation in *S. pombe*, these PY motif-containing antidotes are intrinsically toxic and are kept non-toxic by ubiquitination. To test this idea, we applied the DUB tethering analysis by crossing *S. pombe* strains expressing individually GFP-tagged antidotes of four *S. octosporus wtf* genes (*octo_wtf14*, *octo_wtf21*, *octo_wtf25*, and *octo_wtf46*) to an *S. pombe* strain expressing GBP-DUB. Cross progeny harboring both the *GFP-octo_antidote* transgene and the *GBP-DUB* transgene were inviable (Fig. 6b). These results suggest that ubiquitination-mediated toxicity

neutralization is an ancient mechanism dating back to the common ancestors of *wtf* genes existing 100 million years ago.

## Discussion

Our study uncovers a ubiquitination-mediated toxicity neutralization mechanism employed by *wtf* KMDs to convert the intrinsically toxic long isoform product into a non-toxic antidote (Fig. 6c). This conversion requires conserved PY motifs, which promote ubiquitination through binding Rsp5/NEDD4 family ubiquitin ligases. Ubiquitination serves as a sorting signal to direct the trafficking of the antidote protein from the TGN to the endosome, and thereby prevents it from exerting toxic effects at the TGN and/or at non-endosomal post-TGN locations. The ability of the antidote to interact with the toxin allows the antidote to target the toxin to the endosome and thereby neutralize the toxicity of the toxin.

It remains unclear how non-ubiquitinated Wtf proteins cause toxicity. The partial co-localization of the toxic forms of Cw9 with a TGN marker suggests a possibility that the toxicity is due to perturbation of TGN functions, but we cannot rule out the possibility that toxicity is exerted at non-endosomal post-Golgi membrane locations. It has been proposed that the cytoplasmic puncta formed by the toxin product of *Sk wtf4* are protein aggregates[6]. In this study, we show that Cw9t is a transmembrane protein that traffics through the secretory pathway from the ER to the TGN, where it exhibits a punctate localization pattern, presumably because that is how the TGN appears in light microscopy.

The mechanisms uncovered in this study can serve as a paradigm for understanding how toxin-antidote duality of single-gene KMDs is generated. We propose that for single-gene KMDs, ubiquitination or other types of post-translational modifications may be a commonly employed mechanism for converting a toxic protein product to a non-toxic protein or converting a non-toxic protein product to a toxic protein. As a result, both a toxic protein and a non-toxic protein can be produced from the same gene. An advantage of such a mechanism is that at the amino acid sequence level, post-translational modifications can be determined by simply a short sequence motif such a single PY motif. Interestingly, we noticed that another single-gene KMD family, the *Spk-1* gene and its homologs in *Neurospora*[10], encode transmembrane proteins containing a conserved PY motif (Supplementary Fig. 7b), suggesting the possibility that PY motif-mediated ubiquitination may control the toxin-antidote duality of *Spk-1* family KMDs.

For single-gene KMDs that, like *wtf* driver genes, express overlapping but non-identical protein products, post-translational modification-based toxin-antidote duality can be achieved through the presence of modification determinant sequence motif(s) in the product-specific amino acid sequence. For single-gene KMDs that express a single polypeptide, post-translational modification-based toxin-antidote duality may be realized through variable activities of modifying/demodifying enzymes. For example, a modifying enzyme whose activity is up-regulated in spores may prevent the toxicity of the

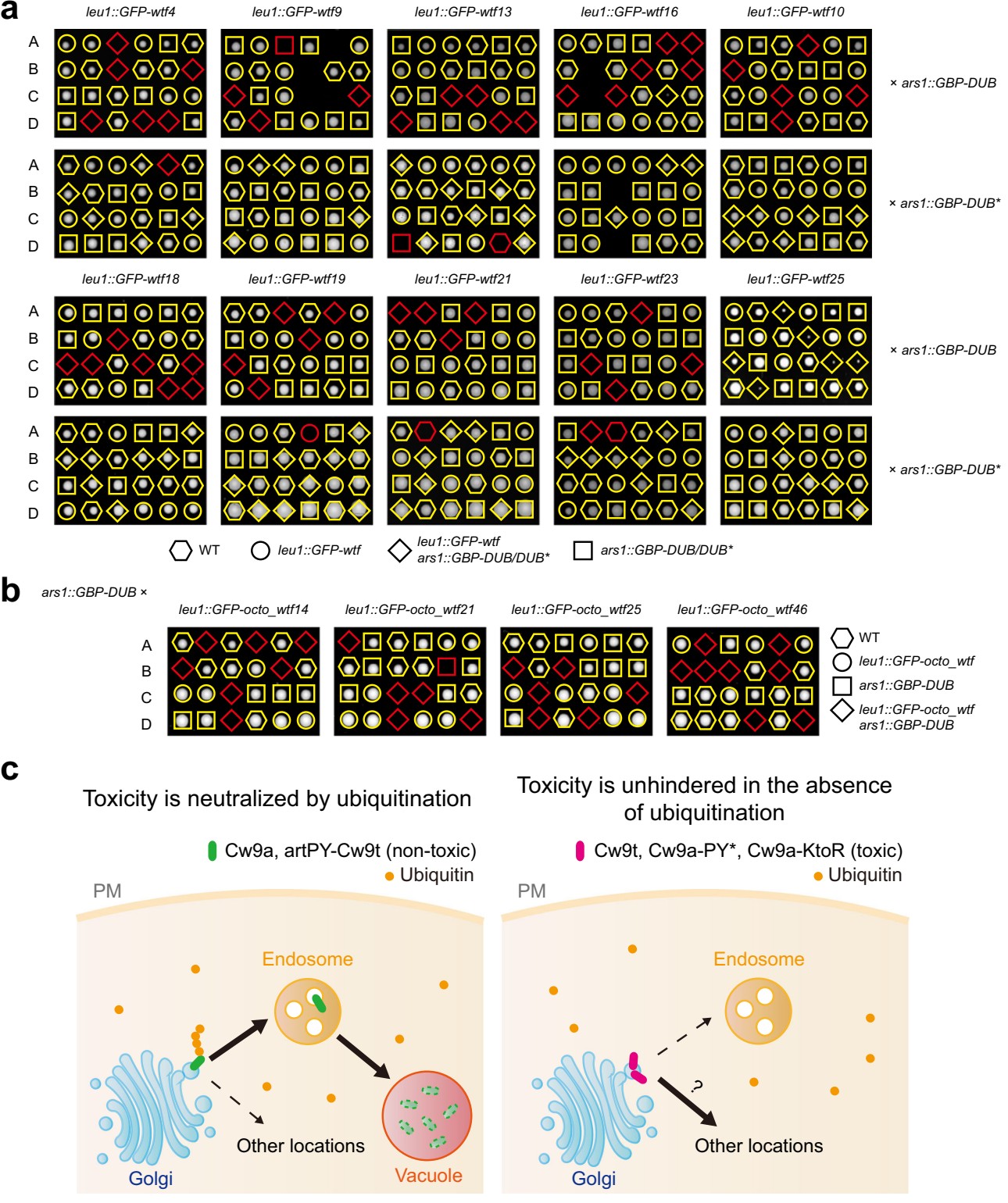

**Fig. 6 | Antidote products of other *wtf* genes are kept non-toxic by ubiquitination.** **a** The antidote products of 10 *wtf* genes of the *S. pombe* reference genome were rendered toxic by artificial DUB tethering. Strains expressing a GFP-fused Wtf antidote protein from the *P41nmt1* promoter were crossed to a strain expressing GBP-DUB or GBP-DUB* from the *P41nmt1* promoter. Mating and sporulation were performed on thiamine-containing media and tetrad dissection was performed on thiamine-free media. Inviable spores whose genotype can be inferred assuming 2:2 segregation are indicated by red symbols. **b** The antidote products of four

*S. octosporus wtf* genes were rendered toxic by artificial DUB tethering. Experiments were performed as in (**a**). **c** A model of the role of PY motif-dependent ubiquitination in determining whether a Wtf protein is a poison or an antidote. PY motif-dependent ubiquitination diverts Wtf proteins from the TGN to the endosome, prevents toxicity, and confers antidote activities. Lack of ubiquitination results in the accumulation of Wtf proteins at the TGN and possibly other additional locations, and allows the manifestation of the toxicity.

newly synthesized KMD product in spores and allow it to detoxify the toxin inherited from the zygote. It is of note that certain eukaryotic toxin-antidote systems encode proteins that may potentially catalyze post-translational modifications of themselves. For example, the protein products of *Spok* spore killers in *Podospora* contain a kinase domain[15], and the *zeel-1* gene of the embryo-killing *peel-1/zeel-1* element in *Caenorhabditis elegans* encodes a protein sharing similarity with a subunit of a ubiquitin ligase complex[50].

For the non-toxic protein product of a single-gene KMD to fulfill the role of the antidote, it needs to exert a toxicity neutralizing activity on the toxin product. As shown here for the case of *cw9*, this need can be satisfied if the toxin and the antidote can interact with each other. As the toxin and antidote products of a single-gene KMD share an overlapping region, the toxin-antidote interaction can be achieved if the overlapping region is capable of self-interaction.

Summing up the above considerations, we propose that toxin-antidote duality of single-gene KMDs can be based on the concurrent presence of the following two features: (1) The gene product(s) can engage in homotypic interactions; and (2) At least one gene product isoform undergoes a post-translational modification that either causes or prevents toxicity. This model has implications for the evolutionary origin of single-gene toxin-antidote KMDs. We envision that any pre-existing genes encoding self-interacting proteins regulated by a post-translational modification are potential precursors of single-gene toxin-antidote KMDs. Should a mutation arise that renders such a protein toxic in either the unmodified or modified form, but not both, a rudimentary KMD may emerge. Subsequent evolutionary processes would then refine its selfish behavior, ultimately culminating in the formation of a fully-developed meiotic driver gene.

## Methods

### Strains and plasmids

*S. pombe* and *S. cerevisiae* strains used in this study are listed in Supplementary Data 2, and plasmids used in this study are listed in Supplementary Data 3. Strain construction methods and media used for *S. pombe* culturing are as described[51,52]. The SC medium (FunGenome, Beijing, China) was used for culturing *S. cerevisiae*. Deletion strains were created using PCR-based gene deletion. Strains expressing proteins fused with various tags from native promoters were created using PCR-based tagging. Plasmids carrying the *Pnmt1*, *P41nmt1*, or *P81nmt1* promoter were based on modified pDUAL vectors[53,54]. The DUB used in the DUB tethering analysis consists of a linker region from the budding yeast Ubp7 (residues 561-591) and the catalytic domain from the herpes simplex virus 1 (HSV-1) UL36 protein (residues 15-260)[35,36]. The catalytically dead DUB* harbors a C40S mutation in the sequence of UL36. The *P81nmt1a* promoter was constructed by inserting a 57 bp sequence that contains the restriction sites of XhoI, SalI, NruI, AscI, PstI, and ApaI between the NheI and BamHI sites downstream of the *P81nmt1* promoter[55]. Plasmids containing the *PtetO7* promoter were constructed by replacing the *Pnmt1* promoter of the pDUAL vector. For the plasmid expressing Sar1-T34N, the coding sequence of Sar1-T34N was inserted into a modified stable integration vector[56]. *cw9t-only* corresponds to the Te form of *cw9* reported in a previous study[9]. Other forms of *cw9* used in this study were generated by PCR-based plasmid construction. Plasmids expressing proteins in budding yeast under the control of the *pGAL1* promoter were constructed using pRS305-based vectors[57]. Plasmids expressing proteins in budding yeast under the control of the *pCYC1*, *pTEF1*, and *pURA3* promoters were constructed using pNH605-based vectors[57]. For the doxycycline (DOX) inducible system in budding yeast, the sequence of *cw9t* was inserted downstream of the *pTETO7* promoter, and the plasmid expressing rtTA-Msn2 was constructed with the pNH607 vector[57]. Plasmids expressing different fragments of Cw9a and truncated Pub1/Pub3 were constructed by inserting coding sequences into pETDuet vectors or pET15b vectors. Plasmids

expressing Uba1, Ubc4, and ubiquitin in *E. coli* were those used in a previous study[58].

### Tetrad analysis

Haploid parental strains freshly grown on YES plates were mixed and spotted on SPAS plates. The SPAS plates were incubated at 30 °C for about 24 h. The tetrad analysis was carried out using a TDM50 tetrad dissection microscope (Micro Video Instruments, Avon, USA) and spores were placed on YES plates. For the DUB tethering analysis, thiamine was added to SPAS plates, and spores were placed on PMG plates without thiamine.

### Spot assay

Log-phase cells grown in the YES medium were washed twice with water and diluted to an $OD_{600}$ of 0.4 in water. 200 μL of cell suspension of each strain was transferred to a well in a 96-well plate. Five-fold serial dilutions of cell suspension were made in the 96-well plate and spotted onto plates using a pin tool (3 mm diameter pins). Plates were incubated at 30 °C for about 2 days. For strains expressing *cw9* from thiamine-repressible promoters, PMG plates with or without thiamine were used. For *S. pombe* strains expressing *cw9* from the *PtetO7* promoter, MSA plates with or without 2.5 μg/mL of anhydrotetracycline (ahTet) (Sigma Aldrich, Cat#37919) were used[59]. For *S. cerevisiae* strains expressing *cw9* from the *pGAL1* promoter, SC plates with 2% galactose or glucose were used. For *S. cerevisiae* strains expressing *cw9* from the *pTETO7* promoter, SC plates with or without 200 μg/mL of doxycycline (DOX) (J&K, Cat#146689) were used.

### Protein extraction

Cell lysates were prepared from ~3 $OD_{600}$ units of cells using a trichloroacetic acid (TCA) (Sigma Aldrich, Cat#T9159) lysis method[60]. The cells were harvested and resuspended in 100 μL of water. Cells were lysed on ice for 15 min by adding 75 μL of NaOH/2-mercaptoethanol (1.85 M NaOH, 7.5% (v/v) 2-mercaptoethanol) to the suspension. The proteins were precipitated by adding 75 μL of 55% (w/v) TCA solution for 10 min on ice. After centrifuging the sample at 15871 × g for 10 min, the supernatant was removed. Following the addition of about 5 μL of neutralizing buffer (1.5 M Tris-HCl, pH 8.8) to prevent any color change, the protein pellet was resuspended in 100 μL of SDS loading buffer (60 mM Tris-HCl, pH 6.8, 4% 2-mercaptoethanol, 4% SDS, 0.01% bromophenol blue, 5% glycerol) and incubated at 65 °C for 20 min. The mouse anti-GFP (Roche, Cat#11814460001, dilution: 1:3000) and horseradish peroxidase (HRP)-conjugated goat anti-mouse IgG (Sigma Aldrich, Cat#A4416, dilution: 1:3000) antibodies were used for detection.

### Immunoprecipitation

About 30 $OD_{600}$ units of cells were harvested. The cells were mixed with 300 μL of lysis buffer (100 mM Tris-HCl, pH 7.5, 50 mM NaCl, 50 mM NaF, 1 mM EDTA, 1 mM EGTA, 10% glycerol, 1% Triton X-100, 1 mM DTT, 1 mM PMSF, 1× Roche protease inhibitor cocktail) and about 500 μL of 0.5-mm-diameter glass beads. Using a Fastprep-24 device, bead beating lysis was carried out. The setting was 6.5 m/s for 30 s followed by a 2-min on-ice cooling. The beating step was repeated three times to make sure >90% of cells were lysed. Microtubes used for bead beating were punctured at the bottom with a hot needle and the lysates were collected into new microtubes by centrifugation at 845 × g for 1 min at 4 °C. The cell lysates were cleared by centrifugation at 15871 × g for 10 min. The supernatants were incubated with GFP-Trap agarose beads (ChromoTek, Cat#gta-20) for 2 h at 4 °C. Following incubation, the beads were washed with lysis buffer devoid of DTT, PMSF, and the protease inhibitor cocktail. Bead-bound proteins were eluted by incubating the beads at room temperature with SDS loading buffer for 10 min. The mouse anti-GFP (Roche, Cat#11814460001, dilution: 1:3000), rabbit anti-mCherry (Abcam, Cat#ab167453,

dilution: 1:3000), mouse anti-FLAG (Sigma Aldrich, Cat#F3165, dilution: 1:3000), HRP-conjugated goat anti-mouse IgG (Sigma Aldrich, Cat#A4416, dilution: 1:3000), and HRP-conjugated goat anti-rabbit IgG (Sigma Aldrich, Cat#A6154, dilution: 1:3000) antibodies were used for detection.

### Affinity purification coupled with mass spectrometry (AP-MS) analysis

AP-MS analysis was performed as described[36] with some modifications. About 1000 $OD_{600}$ units of cells were harvested. Cells were lysed by grinding in liquid nitrogen. The powder was mixed with 30 mL of lysis buffer as described above and incubated for 30 min at 4 °C. After centrifugation, the cell lysate was incubated with GFP-Trap agarose beads for 2 h at 4 °C. After incubation, the beads were washed four times with lysis buffer. Elution was carried out by incubating the beads at 65 °C with SDS loading buffer. Eluted proteins were separated by SDS-PAGE. Protein bands on the SDS-PAGE gel were de-stained and in-gel digested with sequencing grade trypsin (10 ng/μL trypsin, 50 mM ammonium bicarbonate, pH 8.0) overnight at 37 °C. Peptides were extracted with 5% formic acid/50% acetonitrile and 0.1% formic acid/75% acetonitrile sequentially and then concentrated to about 20 μl. The extracted peptides were separated by an pre-column (75 μm × 5 cm) packed with 10 μm spherical C18 reversed phase material (YMC, Kyoyo, Japan) and analytical capillary column (100 μm × 15 cm) packed with 3 μm spherical C18 reversed phase material (Dr. Maisch, GmbH, Germany). A Waters nanoAcquity UPLC system (Waters, Milford, USA) was used to generate the following HPLC gradient: 0-8% B in 10 min, 8-30% B in 30 min, 30-80% B in 15 min, 80% B in 5 min (A = 0.1% formic acid in water, B = 0.1% formic acid in acetonitrile). The eluted peptides were sprayed into a LTQ Orbitrap Velos mass spectrometer (Thermo Fisher Scientific, San Jose, CA, USA) equipped with a nano-ESI ion source. The mass spectrometer was operated in data-dependent mode with one MS scan followed by ten HCD (High-energy Collisional Dissociation) MS/MS scans for each cycle. With the cutoff of peptide FDR < 1%, proteins were identified using the pFind program[61].

### Recombinant protein purification

Recombinant proteins containing the $His_6$ tag were expressed in *E. coli* BL21 or Rosetta strain. *E. coli* cells were cultured in 200 mL of LB medium (10 g/L bactotryptone, 5 g/L yeast extract, 10 g/L NaCl) containing 50 mg/L of ampicillin. When $OD_{600}$ reached 0.4–0.6, 40 μL of 1 M IPTG was added and the culture was incubated at 16 °C for 18 h. The following steps were performed at 4 °C or on ice. The cells were harvested and mixed with 8 mL of *E. coli* lysis buffer (50 mM $NaH_2PO_4$, pH 8.0, 300 mM NaCl, 10 mM imidazole, 10% glycerol). The cell suspension was transferred to a 10 mL beaker, and cells were lysed by sonication. The lysate was centrifuged at 2129 × g for 10 min at 4 °C. The supernatant was incubated with Ni-NTA beads (QIAGEN, Cat#30210) at 4 °C for 1 h. The beads were washed three times with wash buffer (50 mM $NaH_2PO_4$, pH 8.0, 300 mM NaCl, 20 mM imidazole, 10% glycerol). Elution was performed four times by incubating the beads each time with an equal volume of elution buffer (50 mM $NaH_2PO_4$, pH 8.0, 300 mM NaCl, 250 mM imidazole, 10% glycerol). The eluate was stored at −80 °C.

### GST pulldown

For analyzing the interactions between Cw9a(1-52) and three E3 ligases (Pub1, Pub2, and Pub3) and the interactions between Cw9a N-terminal fragments and Pub3, the E3 ligases were expressed in *S. pombe* and Cw9a fragments were expressed in *E. coli*. *S. pombe* lysates were generated the same way as in the immunoprecipitation analysis. *E. coli* lysates were generated the same way as in recombinant protein purification, except for the composition of the lysis buffer (50 mM phosphate buffer, pH 7.4, 200 mM NaCl, 10% glycerol, 0.05% NP-40). After

centrifugation, 1 mL of *E. coli* lysate and 100 μL of *S. pombe* lysate were mixed and incubated at 4 °C for 30 min before the addition of glutathione-sepharose beads (GE Healthcare, Cat#17-0756-01). After 2-h incubation, the beads were washed four times with *E. coli* lysis buffer and subsequently eluted with SDS loading buffer. The mouse anti-GST (BPI, Cat#AbM59001-2H5-PU, dilution: 1:3000), mouse anti-FLAG (Sigma Aldrich, Cat#F3165, dilution: 1:3000), and HRP-conjugated goat anti-mouse IgG (Sigma Aldrich, Cat#A4416, dilution: 1:3000) antibodies were used for detection.

For the interactions between Cw9a N-terminal fragments and Pub1-ΔC2, proteins were all expressed in *E. coli* and the other steps were as described above. The mouse anti-GST (BPI, Cat#AbM59001-2H5-PU, dilution: 1:3000), mouse anti-HA (MBL, Cat#M180-3, dilution: 1:3000), and HRP-conjugated goat anti-mouse IgG (Sigma Aldrich, Cat#A4416, dilution: 1:3000) antibodies were used for detection.

### In vitro ubiquitination

In vitro ubiquitination was performed as described[30] with some modifications. Mix 1 was prepared as follows: 4 μL of buffer A (50 mM Tris-HCl, pH 7.5, 100 mM NaCl, 10% glycerol), 2 μL of 5 × ATP (10 mM ATP, 50 mM $MgCl_2$, 5 mM DTT in buffer A), 1 μL of Uba1 (0.1 μg), 1 μL of Ubc4 (2.5 μg), 2 μL of ubiquitin (7 μg). Mix 1 was preincubated for 10 min at 25 °C. Mix 2 was prepared as follows: 6.5 μL of buffer A, 2 μL of 5 × ATP, 0.5 μL of Pub1 (0.3 μg), 1 μL of Cw9a N-terminal fragment (2 μg) as the substrate protein, which was fused to eight copies of a modified myc tag lacking lysine (myc*: EQRLISEEDL)[62]. Mix 1 was added to Mix 2 and incubated for 3 h at 25 °C. The reaction was stopped by adding SDS loading buffer. During immunoblotting, immediately after the protein transfer step, the PVDF membrane was treated for 30 min with 1% glutaraldehyde in PBS to render the transferred proteins more tightly associated with the membrane[63]. The mouse anti-myc (Huaxingbio, Cat#HX1802, dilution: 1:3000) and the HRP-conjugated goat anti-mouse IgG (Sigma Aldrich, Cat#A4416, dilution: 1:3000) antibodies were used for detection.

### In vitro ubiquitination coupled with mass spectrometry analysis

For in vitro ubiquitination followed by MS analysis, 10 μg of the substrate, 0.5 μg of Uba1, and 0.5 μg of Pub1/Pub3 were used. The ubiquitination reaction was carried out at 25 °C overnight. $CaCl_2$ was added to a final concentration of 1 mM. The samples were digested with trypsin overnight at 37 °C. Formic acid was added to the final concentration of 5% to stop the digestion reaction. LC-MS/MS analysis was performed on an Easy-nLC 1000 II HPLC instrument (Thermo Fisher Scientific, San Jose, CA, USA) coupled to an Orbitrap QE-HF mass spectrometer (Thermo Fisher Scientific, San Jose, CA, USA). Peptides were loaded on a pre-column (100 μm ID, 4 cm long, packed with C18 10 μm 120 Å resin from YMC Co., Ltd) and separated on an analytical column (75 μm ID, 10 cm long, packed with Luna C18 1.8 μm 100 Å resin from Welch Materials) using an acetonitrile gradient from 0% to 30% in 70 min at a flow rate of 250 nL/min. The top 20 intense precursor ions from each full scan (resolution 60000) were isolated for higher-energy collisional dissociation tandem mass spectrometry spectra analysis (HCD MS2; normalized collision energy 27) with a dynamic exclusion time of 45 s. Tandem mass spectrometry fragment ions were detected using orbitrap in a normal scan mode. The pFind software was used to identify proteins with the cutoff of peptide FDR < 1%.

### Streptavidin pulldown of biotin-tagged ubiquitin

About 30 $OD_{600}$ units of *S. pombe* cells were harvested. Cells were resuspended with 500 μL of water and incubated at room temperature for 10 min after the addition of 500 μL of 0.7 M NaOH. After centrifugation, the supernatants were discarded. The pellets were resuspended with 1 mL of lysis buffer (2% SDS, 60 mM Tris-HCl, pH 6.8, 5% glycerol, 4% 2-mercaptoethanol) and incubated at 42 °C for 20 min. After centrifugation, the supernatants were incubated with 20 μL of

high capacity streptavidin agarose (Thermo Fisher, Cat#20359) at room temperature for 3 h. After incubation, the beads were washed two times with wash buffer A (50 mM Tris-HCl, pH 7.5, 150 mM NaCl, 1 mM EDTA, 1 mM EGTA, 1% Triton X-100, 0.4% SDS, 1% NP-40, 1× Roche protease inhibitor cocktail), one time with wash buffer B (50 mM Tris-HCl, pH 7.5, 2% SDS), and two times with wash buffer A. Elution was performed two times by incubating the beads each time with 20 µL of elution buffer (50 mM Tris-HCl, pH 8.0, 2% SDS, 5 mM biotin) at 60 °C for 20 min. The mouse anti-myc (Huaxingbio, Cat#HX1802, dilution: 1:3000) and the HRP-conjugated goat anti-mouse IgG (Sigma Aldrich, Cat#A4416, dilution: 1:3000) antibodies were used for detection.

### Electron microscopy

Approximately 20 $OD_{600}$ units of cells treated with 2.5 µg/mL ahTet for 6 h were harvested, and then washed once with water. Cells was fixed with freshly prepared 1% glutaraldehyde and 4% KMnO4, and then dehydrated through a graded ethanol series and embedded in Spurr's resin[64]. An FEI Tecnai G2 Spirit electron microscope operating at 120 kV with a 4k × 4k Gatan 895 CCD camera was used to examine thin sections.

### Fluorescence microscopy

For imaging of vegetative cells expressing Cw9 from a thiamine-repressible promoter, cells were first cultured in the YES medium, which contains thiamine. Log-phase cells were harvested and washed two times with the EMM medium, which lacks thiamine. Next, cells were cultured in the EMM medium to induce protein expression. For imaging of spores, we crossed the two haploid strains on the SPAS plates. The SPAS plates were incubated at 30 °C for about 24 h. Cells were scraped from the SPAS plates and placed onto slides for imaging.

Images shown in Figs. 1d, 5a, c, e, f, g and Supplementary Figs. 1b, 3f, 4c, d were obtained using a DeltaVision PersonalDV system (Applied Precision) equipped with an mCherry/YFP/CFP filter set (Chroma 89006 set) and a Photometrics Evolve 512 EMCCD camera. The Soft-WoRx program and Photoshop CS6 were used to analyze the images. Images shown in Fig. 5k and Supplementary Figs. 3c, d, g, h, 4b, e, 5b, d, e were obtained using an Andor Dragonfly 200 high speed confocal microscope system equipped with a Sona sCMOS camera. Images were acquired using the Fusion software and analyzed using Fiji and Photoshop CS6. During image analysis, we specifically opted to select cells that exhibited no obvious crater-like depressions in the DIC channel, as those with such features were found to show attenuated fluorescence signal of mECitrine[int]-Cw9t.

### Protein sequence alignment

For the sequence alignment shown in Fig. 3a, the N-terminal cytosolic tails of Cw9a, Cw27pi, and Wtf proteins encoded by the 16 non-pseudo *wtf* genes in the reference *S. pombe* genome were analyzed. The ranges of the N-terminal cytosolic tails were based on the transmembrane topology predicted by TOPCONS (https://topcons.cbr.su.se/)[65] for the four divergent Wtf proteins (Wtf7, Wtf11, Wtf14, and Wtf15), and the transmembrane topology described in our previous study for the other 14 proteins[9]. Sequence alignment was performed using the MAFFT web server (https://mafft.cbrc.jp/alignment/server/) with the L-INS-i algorithm[66,67].

For the sequence alignment shown in Supplementary Fig. 7a, the long isoform protein sequences of 30 non-pseudo *wtf* genes from *S. osmophilus*[49], 48 non-pseudo *wtf* genes from *S. octosporus*[4], and 2 non-pseudo *wtf* genes from *S. cryophilus*[4] were analyzed using MAFFT. The transmembrane helices of these Wtf proteins were predicted using PolyPhobius (http://phobius.sbc.su.se/poly.html)[68]. Only the N-terminal cytosolic tails were shown in Supplementary Fig. 7a. The sequence alignment shown in Supplementary Fig. 7b was based on a published alignment[10].

### Reporting summary

Further information on research design is available in the Nature Portfolio Reporting Summary linked to this article.

## Data availability

The authors declare that all data supporting the findings of this study are available within the paper and its supplementary information files. Source data are provided with this paper.

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

## Acknowledgements

We thank Dan Zhao for providing a plasmid containing the *P81nmt1a* promoter; Yan Ding for assistance with the in vitro ubiquitination assay; Gaihong Cai and She Chen for assistance with the mass spectrometry analysis; and Ping Wei for providing *S. cerevisiae* plasmids. This work was supported by intramural funding from the National Institute of Biological Sciences, Beijing, and the Tsinghua Institute of Multi-disciplinary Biomedical Research, Tsinghua University to L.-L.D.

## Author contributions

Conceptualization: J.-X.Z. and L.-L.D.; methodology and investigation: J.-X.Z., T.-Y.D., G.-C.S., Z.-H.M., Z.-D.J., W.Hu., F.S., W.He., M.-Q.D., L.-L.D.; writing—original draft: J.-X.Z. and L.-L.D.; writing—review and editing: J.-X.Z. and L.-L.D.

## Competing interests

The authors declare no competing interests.
