## [Peer Review File · Nature Communications]

Ubiquitination-mediated Golgi-to-endosome sorting determines the toxin-antidote duality of fission yeast wtf meiotic driversREVIEWER COMMENTS

Reviewer #1 (Remarks to the Author):

In the manuscript "Ubiquitination-mediated Golgi-to-endosome sorting determines the poison-antidote duality of wtf meiotic drivers", Zheng et al. present a very thorough investigation on the molecular mechanism of the "wtf" poison-antidote meiotic drive genes from *Schizosaccharomyces*. The authors have done a fantastic job delving into the precise machinery of this complicated system and have presented the results in a clear and concise manner. They have covered every possible angle I could think of, both in terms of controls as well as follow-up questions. This work moves the field forward in understanding how single genes can act as both poisons and antidotes and provides valuable prospective for other systems. The manuscript is already in a very polished state, so I only have a few minor suggestions that I hope will aid the authors in making this study more approachable for non-experts.

Abstract

Can you explain what PY motifs are?

A bit confusing wording regarding the bound poison. I understand now after reading the results, but perhaps this could be phrased a bit clearer. Maybe a statement explaining that the poison binds the antidote first?

Introduction

Line 39 - Is it always 2 protein products?

Line 60 - As in the Abstract, this sentence is a bit confusing

Results

I'd recommend just Cw9p and Cw9k, unless there is a specific reason, having them both end in "i" makes them hard to distinguish from each other.

Line 126 - as in the whole tail is 105 amino acids long, and the terminal 52 amino acids are absent in the killing isoform? consider rephrasing to clarify (although it's clear from figure 2A, so may not be an issue in a finally formatted version).

Line 183 - What is GST?

Line 185 - Why is the C2 domain important?

Line 188 - introduced mutations?

Line 225 - consider reordering the sentence on inviability to be clearer.

Line 259 - What is E3? Or E1 and E2 on Line 263

Line 474 - I think more fission yeast species are now known to have wtf's

Discussion

Some additional points that the authors could discuss if they feel like it:

Line 526 - Note that reference 14 proposes something along these lines with autophosphorylation

Ubiquitination has been observed in other TA systems, e.g. peel/zeel in *C. elegans*

Methods

Note that tables S1 and S2 are not readable in their PDF versions.

I'm not an expert on most of these methods, but they appear to be presented in a rigorous manner.

Reviewer #2 (Remarks to the Author):

The manuscript 'Ubiquitination-mediated Golgi to endosome sorting determines the poison-antidote duality of wtf meiotic drivers' describes the cell biological and biochemical principles of how killer meiotic drivers of the wtf family operate. This work provides important new insight on how cells inherit information (or not) and how self genetic elements act.

Jin-Xin Zheng and co-workers have provided a very interesting manuscript with high quality data. They demonstrate that Cw9pi (the protecting isoform) uses PY motifs in its N-terminal cytosolic tail to interact with the WW domains of HECT-type ubiquitin ligases Pub1 and Pub3. This modular interaction is required to ubiquitinate seven N-terminal lysine residues in Cw9pi, which direct this protein from the Golgi along the ESCRT mediated MVB pathway into vacuoles for degradation. Interestingly, it is sorting from the Golgi to endosomes, rather than the degradation itself, that is required to protect cells (1) from the toxicity of Cw9pi itself and (2) to neutralize Cw9ki (the killer isoform).

Overall, the data presented in the manuscript is convincing and it coherently explains how Cw9pi is neutralized (I only have minor points regarding this part, please see below). The authors go on to show that that ubiquitination of Cw9pi is also required to target Cw9ki from Golgi into endosomes. This step is required to neutralize the toxicity of Cw9ki. How ubiquitinated Cw9pi targets Cw9ki from Golgi to endosomes, remains unexplained.

Major point:

(1) Provide mechanistic insight that helps to explain how the ubiquitination of Cw9pi targets Cw9ki from Golgi to endosomes. Do they form some sort of transport complex? Or is it a more indirect mechanism (ubiquitination in trans, or some other indirect effect that changes Cw9ki localization of protein stability?)

Minor points:

- (1) Fig.2.: What are the other interaction partners identified in the pull-down experiment?
- (2) Fig.3.B.: What are the different bands detected in the lanes of the different GST fusion proteins?
- (3) What is the half-life time of Cw9ki with or without Cw9pi (and the PY1*-PY3* or the K7R mutant).

Reviewer #3 (Remarks to the Author):

In this manuscript, the authors investigated a very interesting question about how the poison (Cw9ki) and antidote (Cw9pi) products from the same gene function oppositely. Using a combination of biochemistry, genetics, and cell biology approaches, they found that the antidote can bind to Rsp5/NEDD4 family ubiquitin ligases, Pub1 and Pub3, through the PY motif in its specific region. After ubiquitination, the Cw9ki-Cw9pi complex is transported from TGN to the endosome, eventually internalized by the ESCRT machinery and delivered to vacuole for degradation, which leads to the toxicity neutralization. Overall, the experiments are well carried out, the data is convincing, the necessary controls are in place, and the conclusions are well aligned with the experiments. For improvements of this manuscript, I only have the following suggestions/comments:

1. The interaction between Cw9pi and the E3 was done only by in vitro pull down, the authors should back it up using in vivo co-IP assay.
2. Does ubiquitination (3PY or 7R mutant) status of Cw9pi affect its interaction/co-localization with Cw9ki?

3. The authors should rule out whether Cw9pi can also take AP3 pathway to reach the vacuole. In addition, it will be beneficial to show a second ESCRT component deletion will cause a similar result in blocking the vacuolar degradation.

Reviewer #1 (Remarks to the Author):

In the manuscript "Ubiquitination-mediated Golgi-to-endosome sorting determines the poison-antidote duality of wtf meiotic drivers", Zheng et al. present a very thorough investigation on the molecular mechanism of the "wtf" poison-antidote meiotic drive genes from *Schizosaccharomyces*. The authors have done a fantastic job delving into the precise machinery of this complicated system and have presented the results in a clear and concise manner. They have covered every possible angle I could think of, both in terms of controls as well as follow-up questions. This work moves the field forward in understanding how single genes can act as both poisons and antidotes and provides valuable perspective for other systems. The manuscript is already in a very polished state, so I only have a few minor suggestions that I hope will aid the authors in making this study more approachable for non-experts.

Abstract

Can you explain what PY motifs are?

Response: We thank the reviewer for the positive evaluation of our manuscript. We have revised the sentence in the abstract as follows: "We find that the antidote employs N-terminal PY motifs (Leu/Pro-Pro-X-Tyr) to bind Rsp5/NEDD4 family ubiquitin ligases, which ubiquitinate the antidote."

A bit confusing wording regarding the bound poison. I understand now after reading the results, but perhaps this could be phrased a bit clearer. Maybe a statement explaining that the poison binds the antidote first?

Response: We have followed the reviewer's suggestion and rephrased the sentences. The relevant part of the abstract now reads: "Ubiquitination promotes the transport of the antidote from the trans-Golgi network to the endosome, thereby preventing it from causing toxicity. A physical interaction between the antidote and the toxin enables the ubiquitinated antidote to translocate the toxin to the endosome and neutralize its toxicity."

Introduction

Line 39 - Is it always 2 protein products?

Response: There are cases where more than two protein products are expressed. For example, the S5 locus in rice expresses three protein products: ORF3, ORF4, and ORF5. ORF3 is the antidote, whereas ORF4 and ORF5 together constitute the toxin. We have revised the sentence. It now reads: "The protein products of a toxin-antidote KMD form a toxin-antidote pair."

Line 60 - As in the Abstract, this sentence is a bit confusing

Response: We have revised the text. It now reads: "Ubiquitination directs the

transport of the antidote from the trans-Golgi network (TGN) to the endosome, preventing the antidote from causing toxicity. Furthermore, through a physical interaction between the antidote and the toxin, the ubiquitinated antidote relocalizes the toxin to the endosome and thereby detoxifies the toxin.”

Results

I'd recommend just Cw9p and Cw9k, unless there is a specific reason, having them both end in "i" makes them hard to distinguish from each other.

Response: The lowercase letter "p" is often appended to the end of a protein name to indicate that the name represents a protein rather than a gene. Thus, we have chosen to use “Cw9pi” and “Cw9ki” rather than “Cw9p” and “Cw9k”. But we agree with the reviewer that “pi” and “ki” are not ideal. In the revised manuscript, we have renamed “Cw9pi” to “Cw9a” and “Cw9ki” to “Cw9t”, where "a" and "t" denote "antidote" and "toxin", respectively. In accordance, we have changed the term "poison" to "toxin" throughout the text.

Line 126 - as in the whole tail is 105 amino acids long, and the terminal 52 amino acids are absent in the killing isoform? consider rephrasing to clarify (although it's clear from figure 2A, so may not be an issue in a finally formatted version).

Response: We have rephrased the sentence. It now reads: “The N-terminal cytosolic tail of Cw9a is predicted to be 105 amino acids long, with its most N-terminal 52 amino acids being absent in Cw9t (Fig. 2a).”

Line 183 - What is GST?

Response: In the revised manuscript, we have included the full name of GST (glutathione S-transferase) at its first mention in an earlier subsection.

Line 185 - Why is the C2 domain important?

Response: We have revised the sentence to include a description of the role of the C2 domain. It now reads: “we performed GST pulldown analysis using *E. coli*-expressed Cw9a N-terminal fragments fused with GST and *E. coli*-expressed Pub1 lacking the N-terminal lipid-binding C2 domain but containing all three PY-motif-binding WW domains (Pub1- Δ C2)”. In addition, we have added an explanation on why we used the C2-truncated Pub1 instead of the full-length Pub1: “The removal of the C2 domain has been shown to reduce the aggregation tendency of recombinant Rsp5 protein, while not impacting its ability to ubiquitinate PY-containing substrates³⁰.”

Line 188 - introduced mutations?

Response: We have rephrased the sentence. It now reads: “we introduced tyrosine-to-alanine mutations (denoted by asterisks) into the PY motifs of Cw9a.”

Line 225 - consider reordering the sentence on inviability to be clearer.

Response: We have revised the sentence. It now reads: “When the strain containing both *cw9a-3PY and *cw9a* was crossed to a strain without any forms of *cw9*, the resulting spores that contained only *cw9a-3PY** were inviable, while the viability of other spores was normal (Fig. 3d).”**

Line 259 - What is E3? Or E1 and E2 on Line 263

Response: In the revised manuscript, we have included the full name of E1 (ubiquitin-activating enzyme), E2 (ubiquitin-conjugating enzyme), and E3 (ubiquitin ligase).

Line 474 - I think more fission yeast species are now known to have wtf

Response: In the revised manuscript, we have added that the newly identified fission yeast species, *S. linderi*, also harbors *wtf* genes.

Discussion

Some additional points that the authors could discuss if they feel like it:

Line 526 - Note that reference 14 proposes something along these lines with autophosphorylation Ubiquitination has been observed in other TA systems, e.g. peel/zeel in *C. elegans*

Response: In the revised manuscript, we have expanded the discussion to include the possibility that the protein products of KMDs and other TA systems may catalyze post-translational modifications of themselves.

Methods

Note that tables S1 and S2 are not readable in their PDF versions.

I'm not an expert on most of these methods, but they appear to be presented in a rigorous manner.

Response: The strain table and the plasmid table are Excel files and will be presented in the Excel file format when the manuscript is published.

Reviewer #2 (Remarks to the Author)

The manuscript ,Ubiquitination-mediated Golgi to endosome sorting determines the poison-antidote duality of *wtf* meiotic drivers’ describes the cell biological and biochemical principles of

how killer meiotic drivers of the wtf family operate. This work provides important new insight on how cells inherit information (or not) and how self genetic elements act.

Jin-Xin Zheng and co-workers have provided a very interesting manuscript with high quality data. They demonstrate that Cw9pi (the protecting isoforms) uses PY motifs in its N-terminal cytosolic tail to interact with the WW domains of HECT-type ubiquitin ligases Pub1 and Pub3. This modular interaction is required to ubiquitinate seven N-terminal lysine residues in Cw9pi, which direct this protein from the Golgi along the ESCRT mediated MVB pathway into vacuoles for degradation. Interestingly, it is sorting from the Golgi to endosomes, rather than the degradation itself, that is required to protect cells (1) from the toxicity of Cw9pi itself and (2) to neutralize Cw9ki (the killer isoform).

Overall, the data presented in the manuscript is convincing and it coherently explains how Cw9pi is neutralized (I only have minor points regarding this part, please see below). The authors go on to show that that ubiquitination of Cw9pi is also required to target Cw9ki from Golgi into endosomes. This step is required to neutralize the toxicity of Cw9ki. How ubiquitinated Cw9pi targets Cw9ki from Golgi to endosomes, remains unexplained.

Major point:

(1) Provide mechanistic insight that helps to explain how the ubiquitination of Cw9pi targets Cw9ki from Golgi to endosomes. Do they form some sort of transport complex? Or is it a more indirect mechanism (ubiquitination in trans, or some other indirect effect that changes Cw9ki localization of protein stability?)

Response: We thank the reviewer for the positive evaluation of our manuscript and for raising the question on how ubiquitination of the antidote enables it to neutralize the toxin. We show in Fig. 5d that the antidote can physically interact with the toxin. We propose that this interaction allows the ubiquitinated antidote to alter the subcellular localization of the toxin, leading to its neutralization. In response to reviewer's question, we investigated whether ubiquitination of the toxin is required for its detoxification. We found that mutating all cytosol-facing lysines of the toxin did not affect its neutralization by the antidote, suggesting that ubiquitination of the toxin is not necessary for detoxification. These new results are presented in Supplementary Fig. 5f of the revised manuscript.

Minor points:

(1) Fig.2.: What are the other interaction partners identified in the pull-down experiment?

Response: We have added the full list of *S. pombe* proteins identified in the AP-MS experiment as Table S1 in the revised manuscript. While several other potential interactors were identified, we have focused our analysis on Pub1 and its paralogs for this study.

(2) Fig.3.B.: What are the different bands detected in the lanes of the different GST fusion proteins?

Response: The top band in lanes with multiple bands likely represents intact GST fusion proteins, as judged by the apparent molecular weights. The lower band(s) likely arise from proteolytic cleavage events occurring during preparation of the recombinant proteins. We have added this explanation in the figure legend.

(3) What is the half-life time of Cw9ki with or without Cw9pi (and the PY1*-PY3* or the K7R mutant).

Response: We thank the reviewer for the thoughtful suggestion to examine whether the presence of the antidote affects the half-life of the toxin. However, upon attempting to address this experimentally, we encountered difficulties in making a fair comparison between cells that were unharmed by the toxin due to the presence of the antidote and cells that were being killed by the toxin. Specifically, we found that when expressed from the same inducible promoter, the toxin protein was expressed to a notably higher level in the presence of the antidote than in the absence of the antidote. This is likely because protein expression is perturbed when cells are being killed by the toxin. Given the challenges in normalizing toxin expression between these conditions, we decided not to pursue this comparison further.

Reviewer #3 (Remarks to the Author):

In this manuscript, the authors investigated a very interesting question about how the poison (Cw9ki) and antidote (Cw9pi) products from the same gene function oppositely. Using a combination of biochemistry, genetics, and cell biology approaches, they found that the antidote can bind to Rsp5/NEDD4 family ubiquitin ligases, Pub1 and Pub3, through the PY motif in its specific region. After ubiquitination, the Cw9ki-Cw9pi complex is transported from TGN to the endosome, eventually internalized by the ESCRT machinery and delivered to vacuole for degradation, which leads to the toxicity neutralization. Overall, the experiments are well carried out, the data is convincing, the necessary controls are in place, and the conclusions are well aligned with the experiments. For improvements of this manuscript, I only have the following suggestions/comments:

1. The interaction between Cw9pi and the E3 was done only by in vitro pull down, the authors should back it up using in vivo co-IP assay.

Response: We are grateful for the reviewer's positive evaluation and valuable suggestions. In response to the reviewer's suggestion, we have conducted a co-IP experiment. The results of the co-IP experiment, confirming the interactions between the antidote protein and the E3 proteins, are presented in Supplementary

Fig. 1h of the revised manuscript.

2. Does ubiquitination (3PY or 7R mutant) status of Cw9pi affect its interaction/co-localization with Cw9ki?

Response: To address this inquiry, we examined whether the 3PY* mutant of the antidote protein co-localizes with the toxin protein. The newly obtained results, presented in Supplementary Fig. 3f of the revised manuscript, show that the 3PY* mutant of the antidote protein and the toxin protein exhibited co-localization. Unfortunately, the localization of the 7R mutant of the antidote protein could not be meaningfully assessed, as fusing a fluorescent protein to the 7R mutant rendered it non-toxic. This loss of toxicity is possibly due to the fluorescent protein providing lysines that can be ubiquitinated.

3. The authors should rule out whether Cw9pi can also take AP3 pathway to reach the vacuole. In addition, it will be beneficial to show a second ESCRT component deletion will cause a similar result in blocking the vacuolar degradation.

Response: We examined the localization of the antidote protein in AP3 mutants and found that the vacuole lumen localization of the antidote protein was not affected. These new results are presented in Supplementary Fig. 3h of the revised manuscript.

In addition, we examined the localization of the antidote protein in a second ESCRT mutant, *sst4Δ*, which lacks a component of the ESCRT-0 complex. The phenotype is the same as the *vps24Δ* mutant, with the antidote protein no longer localizing to the vacuole lumen, but instead forming cytoplasmic puncta. These new results are presented in Supplementary Fig. 3g of the revised manuscript.

REVIEWERS' COMMENTS

Reviewer #2 (Remarks to the Author):

The authors have done a thorough revision and have adequately addressed my major concerns.

Reviewer #3 (Remarks to the Author):

In the revised manuscript, the authors have addressed all my comments, and now it is ready for acceptance.